# Escaping the Diversity Trap in Robotic Manipulation via Anchor-Centric Adaptation

Yanzhe Chen [1]   Kevin Yuchen Ma [1 2]   Qi Lv [1]   Yiqi Lin [1]   Zechen Bai [1]   Chen Gao [1]   Mike Zheng Shou [1]

## Abstract

While Vision-Language-Action (VLA) models offer broad general capabilities, deploying them on specific hardware requires real-world adaptation to bridge the embodiment gap. Since robot demonstrations are costly, this adaptation must often occur under a strict data budget. In this work, we identify a critical **diversity trap**: the standard heuristic of "maximizing coverage" by collecting diverse, single-shot demonstrations can be self-defeating due to non-vanishing estimation noise. We formalize this phenomenon as a **Coverage–Density Trade-off**. By decomposing the policy error into estimation (density) and extrapolation (coverage) terms, we characterize an interior optimal allocation of unique conditions for a fixed budget. Guided by this analysis, we propose **Anchor-Centric Adaptation (ACA)**, a two-stage framework that first stabilizes a policy skeleton through repeated demonstrations at core anchors, then selectively expands coverage to high-risk boundaries via teacher-forced error mining and constrained residual updates. Real-robot experiments validate our trade-off framework and demonstrate that ACA significantly improves task reliability and success rates over standard diverse sampling strategies under the same budget.

## 1. Introduction

> *I fear not the man who has practiced 10,000 kicks once, but I fear the man who has practiced one kick 10,000 times.* – Bruce Lee

Vision–Language–Action (VLA) models have demonstrated broad capabilities from pretraining (Zitkovich et al., 2023;

[1]Show Lab, National University of Singapore [2]Institude for Information Research, A*STAR. Correspondence to: Mike Zheng Shou <mike.zheng.shou@gmail.com>.

*Proceedings of the $43^{rd}$ International Conference on Machine Learning*, Seoul, South Korea. PMLR 306, 2026. Copyright 2026 by the author(s).

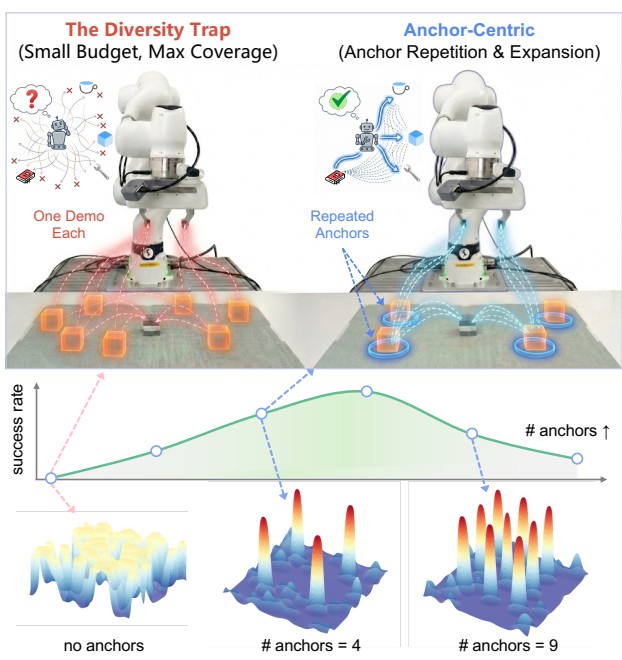

*Figure 1.* **Illustration of Motivation. Top:** Contrasting the "diversity trap" of sparse, single-shot sampling against stable anchor-centric repetition. **Middle & Bottom:** Inverted-U trend of success versus number of anchors, with 3D visualizations of sample distributions and densities at representative points.

Black et al., 2025; Bu et al., 2025). However, deploying these models on specific physical platforms remains a challenge due to the embodiment mismatch and subtle environmental shifts that create substantial distribution gaps (Kawaharazuka et al., 2025; Zhang et al., 2025b). Consequently, real-robot post-training is often a prerequisite for reliable deployment (Xiang et al., 2025; Tan et al., 2025). However, collecting real-world demonstrations is expensive, restricting adaptation to tight data budgets—typically spanning only tens to hundreds of trajectories (Kim et al., 2024; Li et al., 2025a; Chopra et al., 2025). This raises a pivotal question: **Under a limited budget, what is the most effective data collection strategy to learn a robust policy?**

A common approach prioritizes **coverage**—collecting demonstrations across the widest possible array of task conditions to maximize generalizations (Li et al., 2025b;

Hou et al., 2025). While this "maximize coverage" heuristic aligns with standard data-scaling narratives (Liu et al., 2023), we identify it as a **diversity trap** in low-budget regimes. Spreading scarce samples too thin leaves each condition under-represented, thereby inflating estimation variance and destabilizing the resulting vector field. In these scenarios, the fundamental challenge shifts from **what to cover** to **how to allocate**.

Our core insight is that budget-efficient adaptation requires **deliberate repetition before expansion**. This principle echoes the concept of **anchoring**: establishing a sparse set of canonical conditions, or anchors, sampled repeatedly to consolidate a stable "policy skeleton". Empirically, we observe that task success follows an **inverted-U** dependence on the number of anchors. Beyond an optimal point, increasing diversity erodes per-condition density, leading to a catastrophic collapse in policy stability (Fig. 1, bottom).

To formalize this phenomenon, we characterize real-robot post-training as a **Coverage–Density Trade-off**. Under a fixed budget, increasing the number of distinct conditions improves coverage but reduces local sample density, creating a tension between **extrapolation bias** (from unseen regimes) and **estimation noise** (from sparse observations). By analyzing conditional action vector fields (Lipman et al., 2022), we derive a worst-case error bound that explicitly decomposes into these terms. This derivation proves the existence of an **interior optimal allocation**, challenging the optimality of uniform diverse sampling under data scarcity.

Building on this framework, we propose Anchor-Centric Adaptation (ACA), a two-stage framework that navigates this trade-off. **Stage 1 (Anchor-Centric Stabilization)** prioritizes estimation stability by concentrating the budget on a minimal set of anchors, consolidating a low-variance base policy. **Boundary Mining via Teacher-Forced Deviation** then utilizes the base policy's field error as a proxy for extrapolation risk, efficiently identifying under-supported boundary regimes. Finally, **Stage 2 (Constrained Residual Adaptation)** integrates boundary-specific knowledge via a *parameter-efficient residual pathway*. This additive adaptation enables the model to rapidly incorporate critical data from identified boundaries while preventing performance degradation on the previously consolidated core. On real-robot platforms, ACA significantly improves adaptation robustness and task success, demonstrating the necessity of structured data allocation in the small-budget regime.

Our contributions are summarized as follows:

- **Empirical Insight**: We identify the **diversity trap** in VLA post-training, demonstrating that maximizing coverage under tight budgets destabilizes learning, evidenced by a characteristic inverted-U performance trend.

- **Theoretical Formalization**: We introduce the **Coverage–Density Trade-off** framework, providing an analytical bound that reveals the optimal sampling density required to balance estimation and extrapolation bias.

- **Practical Framework**: We propose **ACA**, a two-stage approach employing error-driven boundary acquisition and constrained residual adaptation, improving task success in data-scarce real-robot experiments.

## 2. Related Work

### 2.1. Vision-Language-Action (VLA) Models

VLA models unify perception, instruction following, and control by coupling large-scale vision–language pretraining with action prediction, yielding strong semantic generalization and zero-/few-shot transfer in robotics settings (Zitkovich et al., 2023; Bjorck et al., 2025; Li et al., 2024a; Black et al., 2024; Kawaharazuka et al., 2025; Mu et al., 2023; Ahn et al., 2022; Chen et al., 2025). A prominent line treats actions as tokens within an autoregressive multimodal model, enabling web-scale knowledge to shape robot behavior (Kim et al., 2024; Cen et al., 2025b; Pertsch et al., 2025; Zhang et al., 2026). More recent VLAs move toward continuous-time / high-rate control by generating action trajectories with diffusion- or flow-based policies, improving smoothness and temporal fidelity while retaining the benefits of large-scale pretraining (Black et al., 2025; Zhang et al., 2025d; Cen et al., 2025a; Jiang et al., 2025; Shukor et al., 2025; Zhong et al., 2025). Large cross-embodiment datasets and open-source generalist policies further accelerate VLA scaling, evaluation, and downstream adaptation (O'Neill et al., 2024; Zhang et al., 2025a; Li et al., 2024b; Team et al., 2025; Fei et al., 2025).

### 2.2. Real-World VLA Adaptation

Adapting pretrained VLAs to a new scene typically requires in-domain trajectories to bridge embodiment mismatch and subtle distribution shift, motivating data-efficient post-training pipelines (Kim et al., 2025; Team et al., 2024; Lin et al., 2025; Lee et al., 2025; Kachaev et al., 2025; Li et al., 2025c; Zhao et al., 2025; Mark et al., 2024). A standard heuristic is to maximize condition diversity, but small budgets amplify optimization variance and can render fine-tuning brittle when demonstrations are sparse per condition (Sridhar et al., 2025; Yang et al., 2025). Recent adaptation methods explore parameter-efficient updates, modularity, and constrained refinement to improve robustness (Zhang et al., 2025e; Ni et al., 2025; Wen et al., 2025; Zhang et al., 2025c). Parallel work views adaptation through the lens of continual learning, seeking to prevent catastrophic forgetting while acquiring new skills (Römer et al., 2026; Garcia et al., 2025; Zheng et al., 2025; Hafez & Wermter, 2023). However, most of these approaches are

agnostic to the *acquisition strategy* of the new data itself. Active or error-driven data collection has also been investigated to prioritize demonstrations that provide the highest marginal value for downstream success under limited budgets (Zhu et al., 2025; Yu et al., 2025; Xue et al., 2025; Wu et al., 2025; Gao et al., 2025; Wang et al., 2023; Zhu et al., 2022; Wang et al., 2024). We introduce a *Coverage–Density trade-off* that explains when "maximize coverage" fails. We bridge this gap with ACA to ensure policy skeleton stability before expanding coverage to high-risk regimes.

## 3. Coverage–Density Trade-off

This section formalizes the **Coverage–Density Trade-off** inherent in real-robot adaptation under tight data budgets. We analyze the adaptation process as learning a time-dependent conditional vector field via flow-matching (Lipman et al., 2022). Our analysis yields: (i) an error bound decomposing into *density (estimation)* and *coverage (extrapolation)* terms; (ii) an interior optimal allocation $K^\star$ that identifies the suboptimality of fully diverse sampling; and (iii) a theoretical foundation for the ACA pipeline. Below we outline the key steps; the complete analysis is provided in § A.1.

### 3.1. Problem Formulation

Let $p \in \mathcal{P} \subset \mathbb{R}^d$ parameterize geometric variations (e.g., object pose) and $t \in [0, 1]$ denote flow time. The training target is a conditional vector field $f^\star(z, p, t)$, where $z \in \mathcal{Z}$ is the action-state. We consider post-training under a *fixed and small* total budget of $N$ samples / trajectories collected at conditions $\{p\}$. A sampling strategy specifies (i) a set of $K$ distinct conditions (**coverage**) and (ii) the number of repeats per condition (**density**) $\{n_i\}_{i=1}^K$ with $\sum_{i=1}^K n_i = N$. Abstracting the flow-matching objective, we consider the regression target:

$$y = f^\star(z, p, t) + \varepsilon, \quad \varepsilon \sim \text{sub-Gaussian}(\sigma^2), \quad (1)$$

where $\varepsilon$ represents effective noise due to stochasticity and unmodeled effects in real-robot adaptation.

### 3.2. Assumptions

To make the allocation trade-off explicit, we analyze an idealized **nearest-anchor surrogate**. While deep neural policies are globally parameterized, under small budgets the learning signal is primarily supported by the vicinities of observed conditions, motivating a locality-based analysis.

**A1 (Smoothness in Condition).** For all $(z, t)$, $f^\star(z, \cdot, t)$ is $L$-Lipschitz on $\mathcal{P}$:

$$\|f^\star(z, p, t) - f^\star(z, p', t)\| \le L\|p - p'\|. \quad (2)$$

**A2 (Effective Local Estimation).** For fixed $(z, t)$, the effective estimation error at an observed condition $p_i$ de-

creases with its repeat count $n_i$:

$$\mathbb{E}\|\hat{f}(z, p_i, t) - f^\star(z, p_i, t)\| \le \frac{C\sigma}{\sqrt{n_i}}, \quad (3)$$

where $C > 0$ is a universal constant. This abstracts the empirically observed variance reduction from repeated supervision; global parameter sharing may affect constants but preserves the $1/\sqrt{n_i}$ dependence. **Remark.** A2 characterizes the *data-regime* variance-shrinking effect of repeated supervision and is conceptually orthogonal to model capacity: stronger models may improve the constant $C$ but do not remove the $1/\sqrt{n_i}$ dependence on per-condition density.

**A3 (Surrogate Generalization).** For analysis we consider a local predictor that queries the nearest observed condition:

$$\hat{f}(z, p, t) := \hat{f}(z, p_{i(p)}, t), \ i(p) = \arg\min_i \|p - p_i\|. \quad (4)$$

We emphasize that A3 is an *idealized locality surrogate* used to make the coverage term explicit and to expose worst-case dependence on the fill distance $h$. Deep policies may interpolate more smoothly than this surrogate; our claims focus on the *mechanism* of the coverage–density trade-off and the resulting interior optimum, rather than a tight characterization of neural-network generalization. **Remark.** A3 yields a *conservative upper bound* rather than an empirical claim. Smoother interpolators (e.g., neural networks) can only *improve* upon the nearest-anchor surrogate, so the qualitative predictions — interior optimum and two-stage benefit — transfer as lower bounds on achievable performance.

### 3.3. The Coverage–Density Decomposition

Define the *fill distance*

$$h := \sup_{p \in \mathcal{P}} \min_i \|p - p_i\|, \quad (5)$$

representing the worst-case coverage gap in $\mathcal{P}$.

**Proposition 3.1** (Coverage–Density Bound). *Under A1–A3, the worst-case expected field error is bounded by:*

$$\sup_{p,z,t} \mathbb{E}\|\hat{f} - f^\star\| \le \underbrace{\max_i \mathbb{E}\|\hat{f}_i - f_i^\star\|}_{\text{Estimation (Density) Error}} + \underbrace{Lh}_{\text{Extrapolation Bias}},$$
$$(6)$$

*where $\hat{f}_i := \hat{f}(z, p_i, t)$ and $f_i^\star := f^\star(z, p_i, t)$.*

*Proof sketch.* For any $p$, let $p_{i(p)}$ be its nearest anchor. By A3, $\hat{f}(z, p) = \hat{f}(z, p_{i(p)})$. Let $\Delta(z, p) = \|\hat{f}(z, p) - f^\star(z, p)\|$ denote the policy error at point $p$. By the triangle inequality and Assumption A3, the error can be bounded as:

$$\Delta(z, p) \le \Delta(z, p_{i(p)}) + \|f^\star(z, p_{i(p)}) - f^\star(z, p)\|. \quad (7)$$

The second term is bounded by $L\|p - p_{i(p)}\| \le Lh$ via A1. Taking expectation and supremum over $p$ yields Proposition 3.1. $\square$

### 3.4. Scaling Law and the Diversity Trap

To obtain a simple scaling law, we analyze quasi-uniform anchor placement and uniform repeats.

**Lemma 3.2** (Uniform allocation with quasi-uniform anchors). *Assume (i) uniform repeats $n_i = N/K$ for all anchors, and (ii) quasi-uniform anchor placement on $\mathcal{P}$ such that $h \leq cK^{-1/d}$ for some constant $c > 0$. Then*

$$\mathcal{E}(K) := \sup_{p \in \mathcal{P}} \mathbb{E}\|\hat{f} - f^\star\| \ \leq \ C\sigma\sqrt{\frac{K}{N}} + LcK^{-1/d}. \quad (8)$$

*Proof sketch.* By A2 and $n_i = N/K$, $\max_i \mathbb{E}\|\hat{f}(z, p_i, t) - f^\star(z, p_i, t)\| \leq C\sigma\sqrt{K/N}$. By assumption, $h \leq cK^{-1/d}$. Plug into Proposition 3.1. $\square$

**Corollary 3.3** (Interior Optimal Allocation). *Under non-negligible noise $\sigma > 0$, the error $\mathcal{E}(K)$ is minimized at an interior optimal $K^\star$:*

$$K^\star = \left(\frac{2LcN^{1/2}}{dC\sigma}\right)^{\frac{2d}{d+2}} \propto \left(\frac{L^2 N}{\sigma^2}\right)^{\frac{d}{d+2}} < N, \quad (9)$$

*for sufficiently large $N$. At this choice,*

$$\mathcal{E}(K^\star) \ = \ \tilde{O}\left(\sigma^{\frac{2}{d+2}} L^{\frac{d}{d+2}} N^{-\frac{1}{d+2}}\right), \quad (10)$$

*up to constants $(C, c)$.*

**Fully diverse sampling.** The "maximize coverage" heuristic corresponds to $(K = N, n_i = 1)$, giving

$$\mathcal{E}(N) \ \leq \ C\sigma + LcN^{-1/d}, \quad (11)$$

whose leading estimation noise $C\sigma$ **remains constant**. Thus, under non-negligible noise and tight budgets, pushing $K$ toward $N$ is asymptotically suboptimal; repetition is required to reduce estimation error.

### 3.5. Trajectory Error and Two-Stage Motivation

The previous analysis concerns vector-field approximation. To connect it to rollout behavior, let $z(t)$ and $\hat{z}(t)$ denote trajectories induced by $f^\star(\cdot, p, \cdot)$ and $\hat{f}(\cdot, p, \cdot)$ from the same initial condition. If both vector fields are $\Lambda$-Lipschitz in $z$ over the relevant domain and

$$\delta(p) := \sup_{z,t} \|\hat{f}(z, p, t) - f^\star(z, p, t)\|, \quad (12)$$

then Grönwall's inequality gives

$$\|\hat{z}(T) - z(T)\| \leq \frac{e^{\Lambda T} - 1}{\Lambda} \delta(p), \quad (13)$$

with the usual limiting value $T\delta(p)$ when $\Lambda = 0$. Thus field errors can be amplified along a trajectory, making large coverage gaps especially harmful in long-horizon execution. We use this relation as a conservative proxy: it motivates identifying regions where extrapolation-induced field error is likely to translate into large rollout deviation.

The same decomposition also motivates a two-stage adaptation strategy. In an initial stage, repeated demonstrations at a small set of anchors reduce the density term and stabilize the update. In a second stage, additional conditions are selected from boundary or high-deviation regions to reduce the dominant coverage term. Let $\mathcal{P}_{\text{bd}} \subset \mathcal{P}$ denote such boundary regimes and define

$$h_{\text{bd}} := \sup_{p \in \mathcal{P}_{\text{bd}}} \min_i \|p - p_i\|. \quad (14)$$

When $h_{\text{bd}}$ is large, Proposition 3.1 predicts that boundary error is dominated by $Lh_{\text{bd}}$. A second-stage expansion targeted to these regimes directly attacks this term, while parameter-efficient residual updates help limit unnecessary drift on already-stabilized anchors. This does not claim global optimality of a particular neural training procedure; rather, it explains why core repetition followed by selective boundary expansion is the natural budget allocation suggested by the coverage–density trade-off.

## 4. Anchor-Centric Adaptation (ACA)

We introduce **ACA**, a two-stage framework that operationalizes the coverage–density trade-off for budget-efficient VLA adaptation (Fig. 2). ACA comprises three operational steps: **(i) Anchoring** — concentrating the budget on repeated demonstrations at sparse core conditions to stabilize a low-variance policy skeleton; **(ii) Boundary Mining** — using the Stage 1 policy's teacher-forced deviation as a proxy to identify under-supported, high-risk boundary regimes; and **(iii) Coverage** — integrating boundary-specific knowledge via a parameter-efficient residual update, without drifting the consolidated core.

### 4.1. Architectural Preliminaries

ACA builds upon a flow-based VLA architecture (Black et al., 2024; 2025), which decouples perception and action through a pretrained *VLM* and a task-specific *Action Expert*. The policy is trained by flow matching to learn a time-dependent action vector field conditioned on the task condition $p$ and flow time $\tau \in [0, 1]$.

**Theoretical Mapping.** Our analysis in Sec. 3 studies field approximation *pointwise* at fixed action-state and flow time, which makes the coverage–density mechanism explicit. ACA instantiates the resulting core-to-boundary allocation in a practical pipeline.

**Budget Allocation.** Given a strict total budget of $N$ real-robot trajectories, ACA partitions the interaction into three

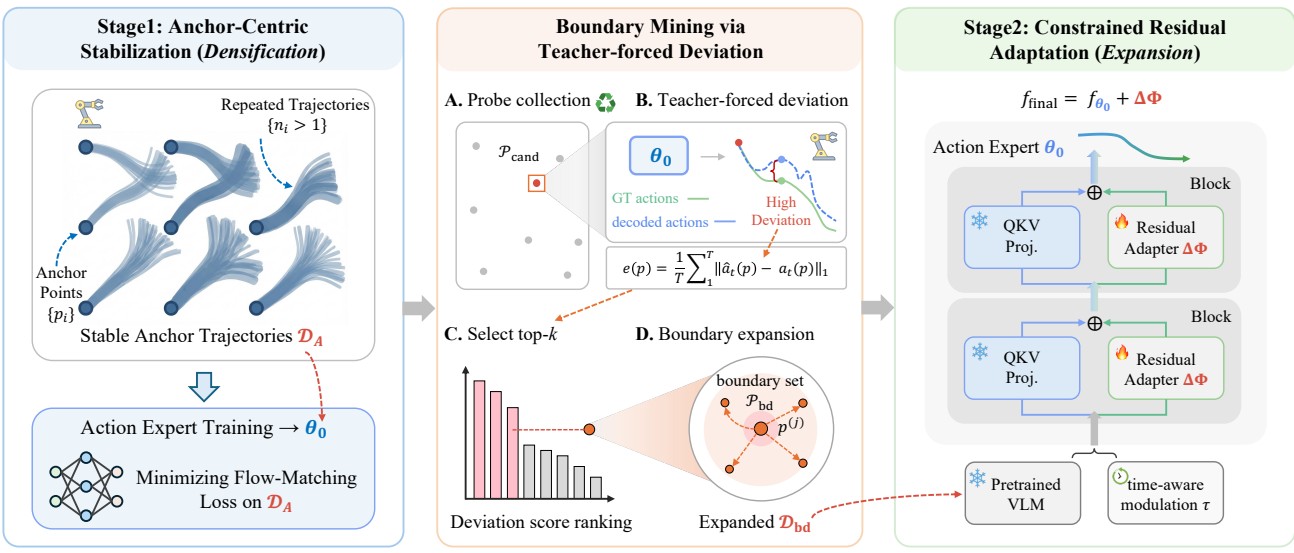

*Figure 2.* **Overview of Anchor-Centric Adaptation (ACA).** ACA comprises three operational steps organized into two training stages. **Stage 1 (Anchoring)** learns a stable core policy by repeating demonstrations at sparse *anchors*; **Boundary Mining** screens probe trajectories via teacher-forced deviation to identify high-risk boundary locations; **Stage 2 (Coverage)** expands boundary competence via a constrained, parameter-efficient residual update in the *Action Expert*, while keeping the pretrained *VLM* frozen.

functional stages:

$$N = N_{\mathrm{A}} + N_{\mathrm{probe}} + N_{\mathrm{bd}}, \qquad (15)$$

where $N_{\mathrm{A}}$ trajectories are dedicated to core anchoring, $N_{\mathrm{probe}}$ trajectories serve as a screening pool for boundary identification, and $N_{\mathrm{bd}}$ trajectories provide the expansion.

### 4.2. Stage 1: Anchor-Centric Stabilization

Stage 1 seeks to reduce *estimation variance* by concentrating the majority of the interaction budget on a sparse set of core anchors $\{p_i\}_{i=1}^{K}$.

**Anchoring Strategy.** Anchors are selected to form a coarse, quasi-uniform cover of the reachable workspace. We collect repeated demonstrations at each anchor ($n_i > 1$), and train a stable core policy $\theta_0$.

**Training.** We train the Action Expert on the anchor dataset $\mathcal{D}_{\mathrm{A}}$ using the standard flow-matching objective:

$$\theta_0 = \arg\min_{\theta} \ \mathbb{E}_{(a,p)\sim\mathcal{D}_{\mathrm{A}}} [\mathcal{L}_{\mathrm{FM}}(\theta; a, p)], \qquad (16)$$

while keeping the *VLM* frozen to leverage pretrained multi-modal representations. This stage yields a policy that is reliable in well-supported regions but may still incur extrapolation error in under-sampled boundary regimes.

### 4.3. Boundary Mining via Teacher-Forced Deviation

After Stage 1 stabilizes the core policy, the dominant remaining error is the *extrapolation* term $Lh_{\mathrm{bd}}$ (Eq. 13) at boundary regimes far from anchors. Boundary mining identifies

these under-supported regimes by leveraging the Stage-1 policy itself as a diagnostic signal — regions where it predicts poorly are exactly where additional supervision yields the highest marginal return.

**Deviation-Based Screening.** We first collect a set of *probe* trajectories by teleoperation at conditions sampled across the workspace, forming a candidate set $\mathcal{P}_{\mathrm{cand}}$ (one probe trajectory per candidate condition). For each candidate condition $p \in \mathcal{P}_{\mathrm{cand}}$, we compute a **teacher-forced deviation score** by decoding actions on the *demonstration* observation sequence and comparing to the demonstrated actions:

$$e(p) := \frac{1}{T} \sum_{t=1}^{T} \|\hat{a}_t(p) - a_t(p)\|_1. \qquad (17)$$

Teacher forcing avoids compounding state drift, making $e(p)$ a clean proxy for action prediction error at condition $p$. By Eq. (13), conditions with high $e(p)$ are precisely those where open-loop trajectory deviation — and hence task failure — is most likely to occur, making $e(p)$ a principled prioritization signal.

**Selecting high-deviation locations.** We select boundary locations using a top-$k$ rule for robustness and simplicity: we take the $k$ candidates with the largest deviation scores. Denote the selected set as $\mathcal{P}_{\mathrm{bd}} = \{p^{(j)}\}_{j=1}^{k}$. Crucially, the corresponding $k$ probe trajectories are *reused* as boundary supervision in Stage 2, so no collected data is wasted.

**Local boundary expansion.** Around each selected boundary location $p^{(j)}$, we sample nearby conditions (e.g., random perturbations within a small neighborhood) and collect ad-

ditional trajectories, forming an expanded boundary dataset $\mathcal{D}_{\text{bd}}$. This targeted expansion selectively reduces $h_{\text{bd}}$ in regimes most detrimental to task success, directly addressing the coverage term identified by our analysis without inflating the budget on already well-supported regions.

### 4.4. Stage 2: Constrained Residual Adaptation

After boundary mining identifies high-risk regions, Stage 2 aims to reduce the coverage gap while preserving the density. Full-parameter fine-tuning is suboptimal here, as boundary gradients may perturb the anchored core. We therefore decouple the two objectives architecturally: the Stage-1 policy is frozen as a stable predictor, and a low-capacity residual branch absorbs boundary-specific corrections.

**Boundary training set.** Stage 2 trains on a boundary dataset $\mathcal{D}_{\text{bd}}$ that combines (i) the reused high-deviation probe trajectories from boundary mining and (ii) the newly collected local-expansion trajectories around $\mathcal{P}_{\text{bd}}$.

**Residual Architecture.** Stage 2 improves boundary behavior *without* drifting the anchored core from Stage 1. We therefore freeze the Stage-1 policy and introduce a small trainable *residual* branch inside the *Action Expert*. For each training instance with condition $p$, observation/context, and flow time $\tau$, the model predicts a time-dependent action vector field at a flow-state $z$. The frozen base policy produces $f_{\theta_0}(z, p, \tau)$, while the residual branch outputs a same-dimensional correction $\Delta_\phi(z, p, \tau)$, yielding

$$f_{\text{final}}(z, p, \tau) = f_{\theta_0}(z, p, \tau) + \Delta_\phi(z, p, \tau), \quad (18)$$

where $\theta_0$ is fixed and only $\phi$ is optimized. We parameterize $\Delta_\phi$ with low-rank adapters (LoRA) (Hu et al., 2022) inserted into selected linear layers of the Action Expert, constraining updates to a low-capacity subspace. Finally, we modulate the residual strength using the flow-time embedding so the correction can specialize across different stages of the denoising process.

**Optimization.** We optimize only $\phi$ on the boundary dataset $\mathcal{D}_{\text{bd}}$ using the same flow-matching objective as Stage 1, while keeping $\theta_0$ frozen. The residual branch is zero-initialized, so optimization starts exactly from the Stage-1 policy. With a fixed base predictor and a low-rank, time-modulated correction, Stage 2 primarily adds localized capacity for boundary regimes while substantially reducing the risk of global drift typical of full-parameter fine-tuning.

## 5. Experiment

We evaluate the ACA framework by addressing three objectives: *(i)* the effectiveness of ACA in mitigating the **diversity trap** and its generalization across VLA backbones; *(ii)* the sensitivity of adaptation to **anchor properties**, specifically the quantity, spatial layout, and consolidation budget $N_A$;

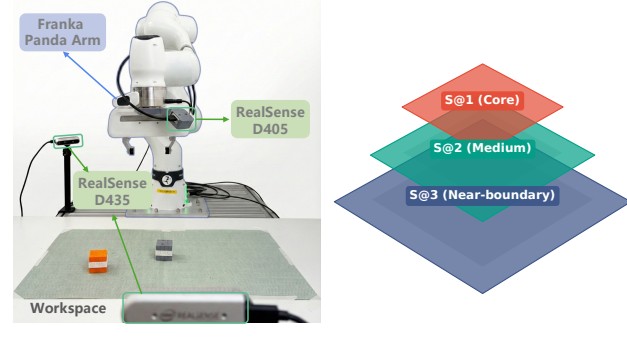

**(a) Real-world Experimental Setup**     **(b) Illustration of S@1- S@3**

*Figure 3.* Illustration of **(a)** the real-robot experimental setup and **(b)** the spatial definition of S@1–S@3 metrics.

and *(iii)* the individual and joint contributions of **boundary mining** and **residual adaptation** to overall performance.

### 5.1. Experimental Setup

We evaluate ACA on real hardware to capture the **non-vanishing estimation noise** ($\sigma > 0$) central to our framework (shown in Sec. 3). Unlike simulation environments that often rely on deterministic transitions or simplified noise models (Peng et al., 2018; Chebotar et al., 2019), real-world interactions exhibit stochastic variability critical for characterizing the "diversity trap". This setting ensures our results reflect the true estimation–extrapolation tension inherent to physical robot adaptation.

**Hardware & Data Collection.** All tasks are performed on a 7-DoF Franka Panda arm. The workspace is monitored by a multi-view camera system, as shown in Fig. 3 (a) comprising a front and rear-side **RealSense D435**, and a wrist-mounted **D405** for localized observation. We collect demonstrations via **leader-follower teleoperation** using two identical Panda arms, providing high-fidelity trajectories for both anchor-centric stabilization and boundary expansion.

**Tasks.** We evaluate ACA on four representative tabletop manipulation tasks that demand high-precision execution across varying workspace regions. *(i) Block Stacking.* Placing a cube onto a target block, requiring the final stack to be stable. *(ii) Cup Placement.* Positioning a cup onto a coaster. *(iii) Table Cleaning.* Using a brush to sweep a block into a dustpan. *(iv) Toy Tidying.* Place toys into a storage box. Real-robot executions of these tasks are shown in Fig. 8.

**Metrics.** To assess robustness against geometric variation, we define **Region-Level Success Rates (S@i)**. Inspired by Recall@$K$ metrics in information retrieval (Radford et al., 2021; Nogueira & Cho, 2019), we categorize the workspace into three nested rectangular regions of increasing difficulty, as illustrated in Fig. 3 (b): **S@1** (core, 25% area), **S@2** (medium, 50% area), and **S@3** (near-boundary, 90% area).

*Table 1.* **Region-level success rates under varying data budgets** $N$. Reported as: *successful trajectories (out of 20 trials) and mean success rate (%).* S@$k$ denotes success within progressively larger central regions (harder as $k$ increases). Budget is partitioned as $N = N_A + N_{probe} + N_{bd}$.

| Method | Block Stacking | | | Cup Placement | | | Table Cleaning | | | Toy Tiding | | | Mean (%) |
|---|---|---|---|---|---|---|---|---|---|---|---|---|---|
| | S@1 | S@2 | S@3 | S@1 | S@2 | S@3 | S@1 | S@2 | S@3 | S@1 | S@2 | S@3 | |
| $N = 50$  ($N_A$=35, $N_{probe}$=5, $N_{bd}$=10) | | | | | | | | | | | | | |
| $\pi_{0.5}$ | 5 | 2 | 0 | 6 | 4 | 0 | 2 | 0 | 0 | 8 | 4 | 2 | 13.8 |
| $\pi_{0.5}$ + **ACA** | 10 | 12 | 9 | 14 | 10 | 10 | 7 | 6 | 5 | 12 | 10 | 6 | 46.3 +32.5 |
| $N = 100$  ($N_A$=70, $N_{probe}$=10, $N_{bd}$=20) | | | | | | | | | | | | | |
| $\pi_{0.5}$ | 8 | 2 | 0 | 14 | 6 | 3 | 10 | 2 | 1 | 14 | 10 | 6 | 31.7 |
| $\pi_{0.5}$ + **ACA** | 16 | 14 | 13 | 18 | 17 | 14 | 12 | 11 | 8 | 18 | 16 | 17 | 72.5 +40.8 |
| $N = 150$  ($N_A$=110, $N_{probe}$=15, $N_{bd}$=25) | | | | | | | | | | | | | |
| $\pi_{0.5}$ | 14 | 8 | 4 | 18 | 13 | 8 | 9 | 7 | 4 | 18 | 14 | 10 | 52.9 |
| $\pi_{0.5}$ + **ACA** | 17 | 18 | 16 | 20 | 20 | 18 | 14 | 11 | 9 | 20 | 19 | 19 | 83.8 +30.9 |

Unless otherwise specified, we sample initial object placements uniformly within each region and report the success rate over 20 evaluation trajectories per task.

**Implementation Details.** We instantiate ACA using flow-matching VLA policies, specifically $\pi_0$ and $\pi_{0.5}$ (Black et al., 2024; 2025). All models are trained using the Adam optimizer with a batch size of 64 on 8 NVIDIA H200 GPUs. Both stages use an action horizon of 16. **Stage 1 (Stabilization).** We train the Action Expert for 20K steps. The learning rate (LR) follows a linear warmup from 0 to $5 \times 10^{-5}$ over the first 1,500 steps, followed by a cosine decay to $2.5 \times 10^{-5}$. **Boundary Mining.** The deviation score $e(p)$ is computed as the $L_1$ norm averaged across 7 dimensions (position, orientation, and gripper state). The number of mined boundary conditions is budget-dependent, with $k \in \{2, 3, 5\}$ corresponding to interaction budgets of $\{50, 100, 150\}$ trajectories, respectively. **Stage 2 (Adaptation).** We perform 10K steps of residual fine-tuning without warmup. The LR starts at $2.5 \times 10^{-5}$ and follows a cosine schedule. We employ LoRA with a rank and $\alpha$ of 32.

### 5.2. Main Results

**Experimental Protocol.** We compare ACA with the baseline on success rates across three spatial regimes (S@1–S@3) under varying data budgets (Table 1). Anchor configuration is fixed to 6 anchors, distributed as the Center Rect type in Fig. 4. The total budget $N$ is partitioned as $N = N_A + N_{probe} + N_{bd}$ following a fixed $\approx$70/10/20 ratio across all settings, as detailed in Table 1. **Baseline Definition.** The baseline ($\pi_{0.5}$) uses standard VLA fine-tuning with maximal-diversity sampling: every demonstration is collected at a distinct, non-repeated condition. **Analysis.** *(i) Consistent gains across data scales*. ACA significantly outperforms the diversity-first baseline at every budget level. For a budget of $N = 100$, ACA achieves a 72.5% mean success rate, yielding a +40.8% absolute improvement over the baseline. *(ii) Alleviating the diversity trap.* The baseline

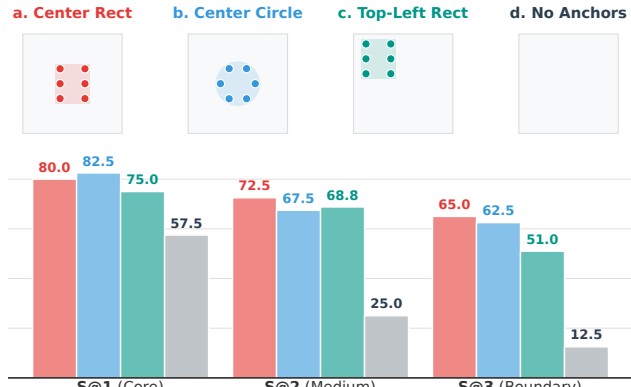

*Figure 4.* **Sensitivity to anchor spatial distribution.** *Top*: Four configurations (a–d) are evaluated under a fixed budget of $N = 100$. *Bottom*: Region-level success rates across the configurations.

frequently collapses to zero successes in near-boundary regions (S@3) due to insufficient per-condition density. ACA alleviates this by first stabilizing the policy skeleton; for example, in Block Stacking at $N = 150$, ACA reaches an 80% success rate (16 / 20) compared to only 20% for the baseline. *(iii) Sample efficiency.* ACA exhibits efficiency gains by prioritizing structured repetition. Notably, ACA at the lowest budget ($N = 50$, 46.3% mean) achieves performance comparable to the baseline at triple the budget ($N = 150$, 52.9% mean). *(iv) Robustness to distribution shifts.* As task conditions move from the central core (S@1) to the boundary (S@3), baseline performance drops precipitously. In contrast, ACA maintains higher stability throughout the entire workspace, confirming that anchor-based consolidation provides a more robust foundation for adaptation.

### 5.3. Ablation Studies

**Anchor Spatial Distribution.** We examine the impact of anchor spatial distribution by fixing the total budget at $N = 100$ trajectories and the number of anchors at $K = 6$,

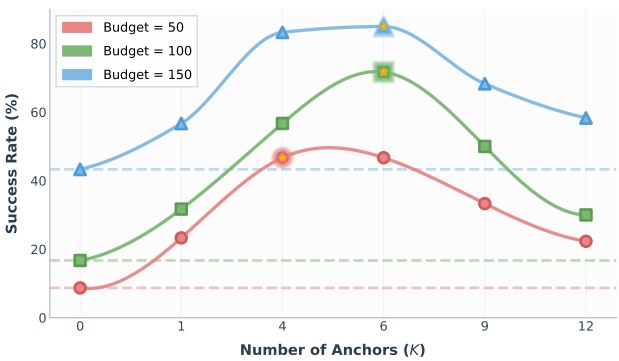

*Figure 5.* **Sensitivity to the number of anchors (K).** Success rates exhibit a consistent inverted-U trend across different budgets.

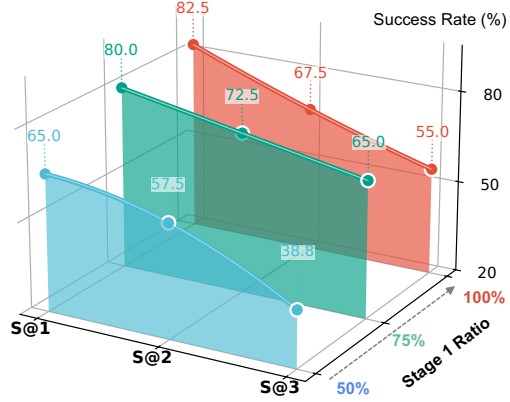

*Figure 6.* **Sensitivity to anchor consolidation budget** ($N_A$). Region-level success rates under a fixed total budget of $N = 100$.

as shown in Fig. 4. *(i) Necessity of anchoring.* All anchor configurations consistently outperform the baseline across all regions. Notably, even the spatially biased top-left layout yields a $4\times$ **improvement** in boundary (S@3) success (51.0% vs. 12.5%), confirming that a stable policy skeleton is more vital than uniform coverage. *(ii) Centralization and reach.* Centralized anchors (a, b) yield the highest success by minimizing the maximum extrapolation distance to any workspace point. This strategy effectively balances local sampling density with global stability, resulting in the most graceful performance degradation. *(iii) Robustness to geometric patterns.* The performance difference between rectangular and circular arrangements appears to be relatively marginal. This result suggests that the benefits of ACA may be largely invariant to the specific geometric pattern, provided that the anchors maintain a sufficiently centralized distribution to influence the surrounding workspace.

**Impact of Anchor Quantity.** We evaluate the impact of anchor quantity by varying $K \in \{0, \dots, 12\}$ under a fixed Center Rect layout, as shown Fig. 5. *(i) Observation of the Inverted-U trend.* Across all tested budgets, task success appears non-monotonic with respect to $K$. This suggests that the benefits of spatial coverage are effectively bottlenecked by the local sampling density required to suppress estimation noise. *(ii) Validation of the Diversity Trap.* As $K$ increases beyond the optimum, success rates tend to decline despite the broader spatial coverage. This supports our hypothesis that excessive diversity under a fixed budget dilutes the sampling density per condition, potentially leading to the "diversity trap" where estimation noise destabilizes the policy. *(iii) Budget-dependent shift in optimum.* The results suggest that the optimal $K$ may scale with the total interaction budget. For instance, the performance peak broadens as $N$ increases from 50 to 150, which is consistent with our theoretical bound predicting that larger budgets can support higher diversity without sacrificing stability.

**Ablation on Boundary Mining and Residual Adaptation.** We fix the total budget at $N = 100$ trajectories with 6 an-

chors, as shown in Table 2. *(i) Synergy of sampling and architecture.* Combining error-driven mining with residual updates yields the highest mean success rate, showing that robust adaptation requires both identifying critical data gaps and a stable update mechanism. *(ii) Effectiveness of error-driven mining.* Adding the mining phase boosts performance in boundary regions (S@3). Compared to a random-sampling baseline of equal size, error-driven mining raises S@3 success, confirming that teacher-forced deviation reliably locates extrapolation risks. *(iii) Stability via residual updates.* While full-parameter fine-tuning allows for boundary expansion, it results in performance trade-offs in the consolidated core. The residual pathway preserves S@1 & S@2 stability while patching the vector field in under-supported areas.

*Table 2.* **Ablation study of Boundary Mining and Residual Adaptation.** Values denote mean success rates across four tasks. The first row is the Stage-1 policy trained only on anchor data ($N_A = 70$); the remaining rows use the full $N = 100$ budget.

| Setting | Mining | Residual | Mean |
|---|---|---|---|
| Stage-1 only ($N_A = 70$) | – | – | 62.1 |
| Random expansion + full FT | × | × | 57.1 |
| Random expansion + residual | × | ✓ | 58.8 |
| Mined expansion + full FT | ✓ | × | 64.2 |
| Full ACA | ✓ | ✓ | **72.5** |

**Ablation on Anchor Budget ($N_A$) Ratio.** We examine the trade-off between anchor consolidation and boundary expansion by varying $N_A$ under a fixed budget of $N = 100$, as shown in Fig. 6. *(i) Stabilization as a prerequisite.* Reducing $N_A$ to 50 results in a global performance decline, with core success (S@1) dropping from 80.0% to 65.0%. This outcome validates that an insufficient anchor budget fails to suppress estimation noise, providing a fragile policy skeleton that cannot effectively support subsequent boundary adaptation. *(ii) Optimal budget partitioning.* While

$N_A = 100$ maximizes central success (82.5%), allocating $N_A = 80$ trajectories yields the highest overall success (72.5% Mean) and superior boundary performance (65.0% vs. 55.0% in S@3). This indicates that once the policy skeleton is consolidated, dedicating a minor portion of the budget to targeted Stage 2 mining provides larger marginal gains than further anchor repetition.

**Robustness to Anchor Placement.** To quantify sensitivity to anchor geometry beyond the four structured layouts in Fig. 4, we evaluate two additional configurations where 6 anchor positions are sampled *uniformly at random* within the workspace ($N = 100$), as shown in Table 3. Both random configurations substantially outperform the no-anchor baseline and closely match the best structured layout (Center Rect, 72.5%). This confirms that **repeated sampling density is the key driver**, rather than precise anchor geometry. In practice, an operator can select a handful of workspace-spanning positions without careful layout engineering, substantially lowering the deployment barrier.

*Table 3.* **Robustness to anchor placement strategy** ($N = 100$, $K = 6$). Mean success rates across four tasks.

| Setting | Structured | Mean (%) |
|---|---|---|
| No Anchors (Baseline) | – | 31.7 |
| Random Placement 1 | × | 70.4 |
| Random Placement 2 | × | 68.8 |
| Center Rect (Fig. 4a) | ✓ | **72.5** |

**Comparison with Data-Collection Baselines.** To contextualize ACA against alternative adaptive collection strategies, we evaluate three additional baselines under identical conditions ($\pi_{0.5}$, $N \in \{50, 100\}$, same budget split). Following each method's standard protocol, all three use diverse single-shot sampling in Stage 1: **Uncertainty-driven** — diverse Stage 1 seed policy; Stage 2 collects at conditions with highest flow-matching variance; **DAgger-style** — diverse Stage 1 seed policy; Stage 2 aggregates human corrections at rollout failure states via replay buffer mixing; **Curriculum** — collects from core regions (S@1–S@2) first, then randomly expands to boundary (S@3). Results are reported in Table 4. All three improve over the naïve baseline, yet ACA remains substantially ahead, with the gap most pronounced at S@3 ($N = 100$: 65.0% vs. ≤27.5%). All three baselines inherit the diversity trap from their single-shot Stage 1, producing an unstable policy skeleton that subsequent expansion cannot rescue. Curriculum shares ACA's core→boundary ordering but lacks **(i)** anchor repetition to suppress estimation noise and **(ii)** deviation-driven boundary selection to target where extrapolation error dominates. ACA is the only method that explicitly addresses both the density and coverage terms in the decomposition of Prop. 3.1.

**Performance Gains across Different VLAs.** We evaluate

*Table 4.* **Comparison with data-collection baselines.** Mean success rates and near-boundary success (S@3) across four tasks.

| Method | Mean (%) | | S@3 (%) |
|---|---|---|---|
| | $N = 50$ | $N = 100$ | $N = 100$ |
| Baseline | 13.8 | 31.7 | 12.5 |
| Uncertainty-driven | 15.0 | 35.4 | 18.8 |
| DAgger-style | 18.3 | 39.2 | 21.3 |
| Curriculum | 23.3 | 42.5 | 27.5 |
| **ACA (ours)** | **46.3** | **72.5** | **65.0** |

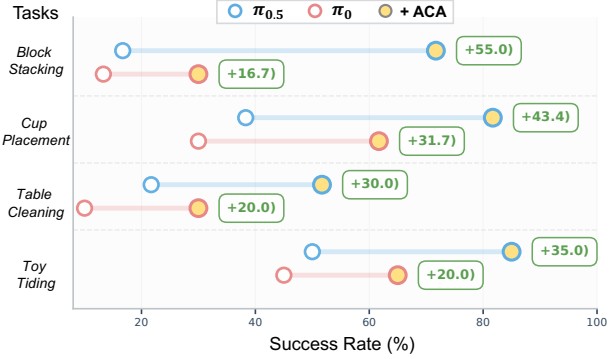

*Figure 7.* **Performance gains across different base VLA models.** ACA yields consistent performance increases on four tasks.

the architectural compatibility of ACA by comparing adaptation gains on $\pi_0$ and $\pi_{0.5}$ backbones, as shown in Fig. 7. **(i)** *Architecture-agnostic efficacy.* ACA improves both models across all tasks, showing that the anchor-centric principle acts as a universal data-allocation strategy for flow-based VLAs. **(ii)** *Synergy with base capabilities.* While ACA enhances both models, the stronger $\pi_{0.5}$ achieves larger absolute gains, indicating that higher initial representation quality enables more effective anchor-based refinement.

## 6. Conclusion

We formalize the **Coverage–Density Trade-off** in VLA adaptation, identifying a **diversity trap** where excessive variety under strict budgets destabilizes learning, evidenced by a characteristic **inverted-U performance trend**. Our framework, **Anchor-Centric Adaptation (ACA)**, operationalizes this insight by establishing a stable policy skeleton at anchors before expanding coverage via error-driven mining and constrained residuals. Real-robot experiments validate that ACA significantly improves task reliability, demonstrating that **structured repetition** is a more potent scaling lever than raw diversity for robust robotic deployment in data-scarce regimes. *Future Work.* Future research includes extending the anchoring principle to long-horizon manipulation via hierarchical decomposition and study autonomous anchor-discovery for resource optimization to scale generalizable skills across broader data distributions.

## Acknowledgements

This research is supported by the National Research Foundation, Singapore under its AI Singapore Programme (AISG Award No: AISG3-RP-2022-030).

## Impact Statement

This work provides a principled framework for the data-efficient adaptation of VLA models, enhancing the reliability and safety of embodied agents in data-scarce environments. By formalizing the Coverage–Density trade-off, ACA mitigates the risks of unpredictable behaviors caused by the diversity trap, ensuring more stable policy skeletons under limited human demonstrations. These contributions support the democratization of robotics, lowering the barrier for deploying high-performance foundation models in specialized, resource-constrained real-world applications.

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

## A. Appendix

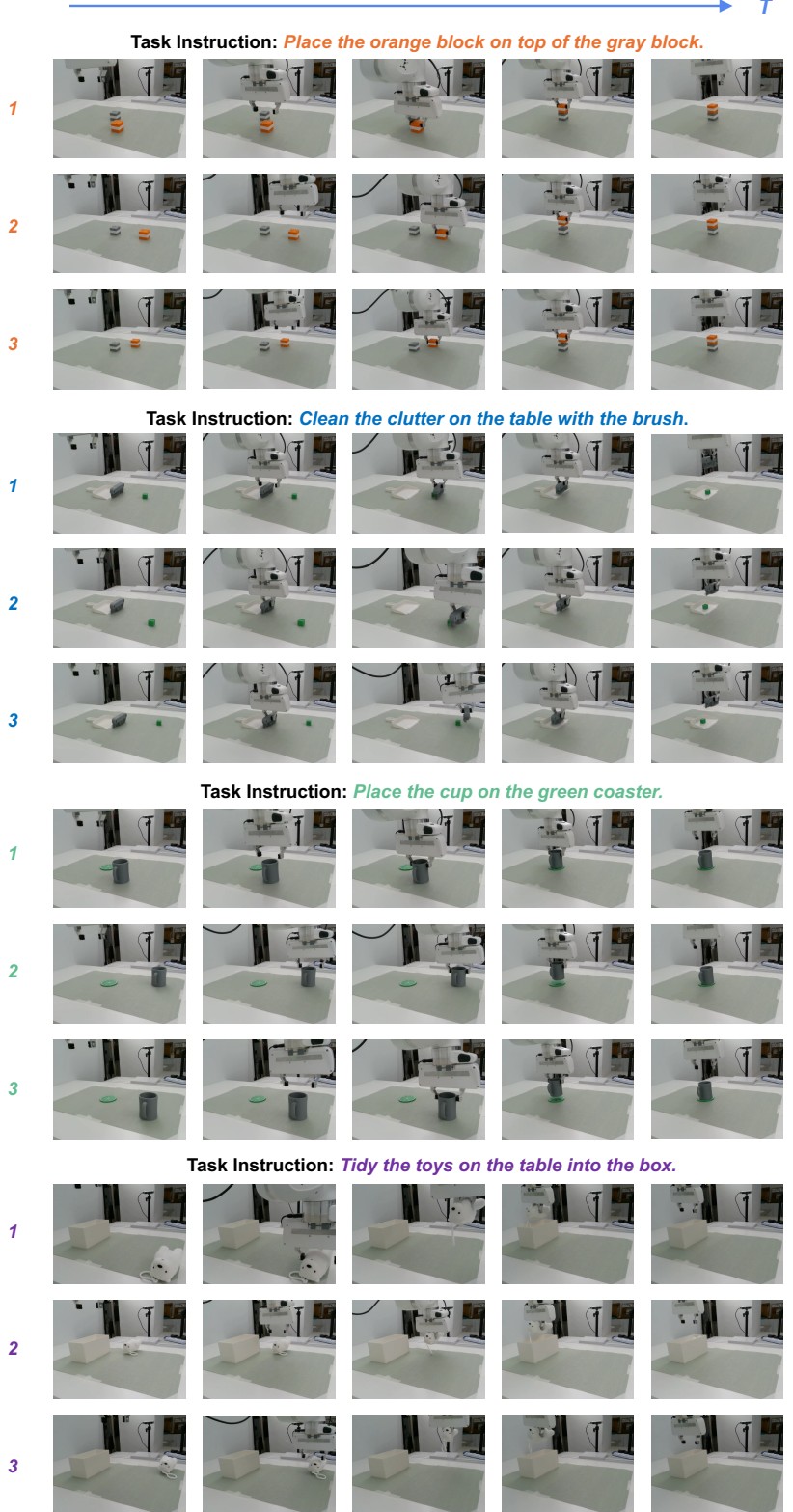

*Figure 8.* Visualization of real-robot rollouts across four tasks: from top to bottom, Block Stacking, Table Cleaning, Cup Placement, and Toy Tidying; each row shows the task instruction and corresponding key frames from the rollout video.

## A.1. Analysis of the Coverage–Density Trade-off

This section provides comprehensive foundations for the Coverage-Density Trade-off presented in Section 3. We present: (i) complete proofs with detailed derivations; (ii) extensions to high-probability bounds; (iii) analysis of non-uniform allocation strategies; (iv) robustness analysis under assumption violations; (v) connections to optimal experimental design; and (vi) sample complexity lower bounds.

### A.1.1. NOTATION AND PRELIMINARIES

We consolidate notation used throughout the theoretical analysis:

*Table 5.* Notation summary for theoretical analysis.

| Symbol | Description |
|---|---|
| $\mathcal{P} \subset \mathbb{R}^d$ | Condition parameter space (dimension $d$) |
| $\mathcal{Z}$ | Action-state space |
| $p, p_i$ | Condition parameters (queries and anchors) |
| $z$ | Action-state variable |
| $t \in [0, 1]$ | Flow time parameter |
| $f^\star(z, p, t)$ | Ground-truth conditional vector field |
| $\hat{f}(z, p, t)$ | Learned/estimated vector field |
| $N$ | Total sample budget |
| $K$ | Number of distinct anchor conditions |
| $n_i$ | Number of repeats at anchor $p_i$ ($\sum_i n_i = N$) |
| $h$ | Fill distance: $\sup_{p \in \mathcal{P}} \min_i \|p - p_i\|$ |
| $L$ | Lipschitz constant of $f^\star$ in $p$ |
| $\Lambda$ | Lipschitz constant of $f^\star$ in $z$ |
| $\sigma$ | Sub-Gaussian parameter of noise |
| $C$ | Universal constant in estimation error bound |
| $c$ | Constant in fill distance bound |
| $\delta$ | Confidence parameter for high-probability bounds |

**Sub-Gaussian Random Variables.** A random variable $X$ is $\sigma$-sub-Gaussian if for all $\lambda \in \mathbb{R}$,

$$\mathbb{E}[e^{\lambda X}] \le e^{\sigma^2 \lambda^2 / 2}. \tag{19}$$

This implies the tail bound $\mathbb{P}(|X| \ge t) \le 2e^{-t^2/(2\sigma^2)}$ and moment bound $\mathbb{E}[|X|^p]^{1/p} \le C_p \sigma$ for constants $C_p$ depending only on $p$.

**Fill Distance and Covering.** For a set of points $\{p_i\}_{i=1}^K \subset \mathcal{P}$, the fill distance quantifies worst-case coverage. For quasi-uniform placement (e.g., grid, low-discrepancy sequences (Niederreiter, 1992)), standard results give $h \asymp K^{-1/d}$ up to logarithmic factors.

## A.2. Theory: Coverage–Density Trade-off

This section formalizes the coverage–density trade-off induced by a finite adaptation budget. The purpose of the analysis is not to claim that a generic neural policy is exactly a nearest-anchor estimator, but to isolate a statistically transparent mechanism: with a fixed number of demonstrations, increasing the number of task anchors improves coverage of the task-parameter space but reduces the number of samples available per anchor. We first prove this trade-off for a nearest-anchor surrogate, then state carefully what changes for stable interpolators, non-uniform allocation, discontinuous task regions, and two-stage sampling.

**Notation.** Let $\mathcal{P} \subset \mathbb{R}^d$ be a compact task-parameter domain equipped with norm $\| \cdot \|$. Let $\{p_i\}_{i=1}^K \subset \mathcal{P}$ be adaptation anchors and define the fill distance

$$h_K := \sup_{p \in \mathcal{P}} \min_{1 \le i \le K} \|p - p_i\|. \tag{20}$$

For each $p \in \mathcal{P}$, let

$$i(p) \in \arg \min_{1 \le i \le K} \|p - p_i\| \tag{21}$$

be an arbitrary nearest-anchor index. The target vector field or policy is denoted by $f^\star(z, p, t)$, where $z$ is the state or conditioning variable and $t \in [0, 1]$ is the flow-matching time variable. The learned predictor is $\hat{f}$. We write $\xi$ for all sampling and optimization randomness used to construct $\hat{f}$. Unless otherwise stated, expectations are taken with respect to $\xi$ and, when explicitly shown, a fixed evaluation distribution $\nu$ over $(z, t)$.

**Assumptions.** We use the following assumptions only where they are explicitly invoked.

**Assumption A.1** (Lipschitz regularity in task parameter). There exists $L < \infty$ such that for all $z$, $t$, and $p, p' \in \mathcal{P}$,

$$\|f^\star(z, p, t) - f^\star(z, p', t)\| \leq L\|p - p'\|. \tag{22}$$

**Assumption A.2** (Nearest-anchor surrogate). For the purpose of the baseline analysis, the predictor at an arbitrary task parameter uses the predictor at its nearest anchor:

$$\hat{f}(z, p, t) = \hat{f}(z, p_{i(p)}, t). \tag{23}$$

Assumption A.2 is deliberately restrictive. It should be read as defining an analyzable surrogate, not as a claim that the neural policy implemented in experiments is a nearest-neighbor rule. Extensions beyond this surrogate require additional stability assumptions and are discussed below.

### A.2.1. COVERAGE–DENSITY DECOMPOSITION

**Proposition A.3** (Coverage–density decomposition). *Suppose Assumptions A.1 and A.2 hold. Then for any evaluation distribution $\nu$ over $(z, t)$,*

$$\sup_{p \in \mathcal{P}} \mathbb{E}_{(z,t) \sim \nu, \xi}\left[\|\hat{f}(z, p, t) - f^\star(z, p, t)\|\right] \leq \max_{1 \leq i \leq K} \mathbb{E}_{(z,t) \sim \nu, \xi}\left[\|\hat{f}(z, p_i, t) - f^\star(z, p_i, t)\|\right] + Lh_K. \tag{24}$$

*The same inequality also holds pointwise for any fixed $(z, t)$ if the expectation over $(z, t) \sim \nu$ is removed.*

*Proof.* Fix any $p \in \mathcal{P}$ and any $(z, t)$. By Assumption A.2,

$$\hat{f}(z, p, t) = \hat{f}(z, p_{i(p)}, t). \tag{25}$$

The triangle inequality gives

$$\|\hat{f}(z, p, t) - f^\star(z, p, t)\| \leq \|\hat{f}(z, p_{i(p)}, t) - f^\star(z, p_{i(p)}, t)\| \tag{26}$$
$$+ \|f^\star(z, p_{i(p)}, t) - f^\star(z, p, t)\|. \tag{27}$$

Assumption A.1 and the definition of $h_K$ imply

$$\|f^\star(z, p_{i(p)}, t) - f^\star(z, p, t)\| \leq L\|p_{i(p)} - p\| \leq Lh_K. \tag{28}$$

Taking expectation over $\xi$ and optionally over $(z, t) \sim \nu$, then taking the supremum over $p$, yields (24). □

**Interpretation.** The first term in (24) is a density term: it decreases when more samples are assigned to each anchor. The second term is a coverage term: it decreases when anchors form a finer cover of $\mathcal{P}$. The trade-off arises because, under a fixed total budget, increasing $K$ improves $h_K$ but reduces the sample count per anchor.

### A.2.2. HIGH-PROBABILITY ANCHOR BOUND

The next result gives a high-probability version under an explicit repeated-query observation model. This model is intentionally stated narrowly: it applies to fixed $(z, t)$ or to settings where anchor errors are already controlled uniformly over the relevant evaluation set. It should not be interpreted as a uniform guarantee over a continuous state-time space without an additional covering or function-class argument.

**Assumption A.4** (Sub-Gaussian anchor observations). For a fixed $(z, t)$ and each anchor $p_i$, suppose we observe

$$y_{ij} = f^\star(z, p_i, t) + \varepsilon_{ij}, \qquad j = 1, \ldots, n_i, \tag{29}$$

where $\varepsilon_{ij} \in \mathbb{R}^m$ are independent, mean-zero, and coordinate-wise $\sigma_i$-sub-Gaussian. The anchor estimate is the sample mean

$$\hat{f}(z, p_i, t) = \frac{1}{n_i} \sum_{j=1}^{n_i} y_{ij}. \tag{30}$$

**Lemma A.5** (Simultaneous concentration at anchors). *Under Assumption A.4, for any $\delta \in (0, 1)$, with probability at least $1 - \delta$, simultaneously for all anchors $i = 1, \ldots, K$,*

$$\|\hat{f}(z, p_i, t) - f^\star(z, p_i, t)\|_2 \le \sigma_i \sqrt{\frac{2m \log(2mK/\delta)}{n_i}}. \tag{31}$$

*For scalar outputs $(m = 1)$, this reduces to*

$$|\hat{f}(z, p_i, t) - f^\star(z, p_i, t)| \le \sigma_i \sqrt{\frac{2 \log(2K/\delta)}{n_i}}. \tag{32}$$

*Proof.* For any fixed anchor $i$ and coordinate $r \in \{1, \ldots, m\}$, the empirical mean error is $\sigma_i / \sqrt{n_i}$-sub-Gaussian. Hence

$$\mathbb{P}\left(|\hat{f}_r(z, p_i, t) - f_r^\star(z, p_i, t)| \ge \sigma_i \sqrt{\frac{2 \log(2mK/\delta)}{n_i}}\right) \le \frac{\delta}{mK}. \tag{33}$$

A union bound over all $mK$ coordinate-anchor pairs gives the coordinate-wise bound simultaneously. The Euclidean bound follows from $\|u\|_2 \le \sqrt{m}\|u\|_\infty$. $\qquad\square$

**Theorem A.6** (High-probability coverage–density bound). *Suppose Assumptions A.1, A.2, and A.4 hold for a fixed $(z, t)$. Let $n_{\min} := \min_i n_i$ and $\sigma_{\max} := \max_i \sigma_i$. Then with probability at least $1 - \delta$,*

$$\sup_{p \in \mathcal{P}} \|\hat{f}(z, p, t) - f^\star(z, p, t)\|_2 \le \sigma_{\max} \sqrt{\frac{2m \log(2mK/\delta)}{n_{\min}}} + Lh_K. \tag{34}$$

*Proof.* On the event of Lemma A.5, every anchor error is bounded by the first term in (34). Applying the deterministic triangle-inequality argument from Proposition A.3 gives the result. $\qquad\square$

### A.2.3. OPTIMAL NUMBER OF ANCHORS UNDER UNIFORM ALLOCATION

Assume the anchors are quasi-uniform in the sense that for a geometry-dependent constant $c_\mathcal{P}$,

$$h_K \le c_\mathcal{P} K^{-1/d}. \tag{35}$$

Assume further that a total budget $N$ is split uniformly across anchors, so $n_i = N/K$, and that the expected anchor estimation error satisfies

$$\max_i \mathbb{E}\|\hat{f}(z, p_i, t) - f^\star(z, p_i, t)\| \le C\sigma \sqrt{\frac{K}{N}}. \tag{36}$$

Combining (24), (35), and (36) gives

$$\mathcal{E}(K) \le C\sigma \sqrt{\frac{K}{N}} + Lc_\mathcal{P} K^{-1/d}. \tag{37}$$

The first term increases with $K$ because each anchor receives fewer samples; the second term decreases with $K$ because the anchor set covers $\mathcal{P}$ more finely.

Ignoring integer constraints, minimizing the right-hand side of (37) yields

$$K^\star \asymp \left( \frac{L^2 c_{\mathcal{P}}^2 N}{\sigma^2} \right)^{\frac{d}{d+2}}, \tag{38}$$

and the optimized error scales as

$$\mathcal{E}(K^\star) \asymp (C\sigma)^{\frac{2}{d+2}} (Lc_{\mathcal{P}})^{\frac{d}{d+2}} N^{-\frac{1}{d+2}}. \tag{39}$$

With the high-probability bound in Theorem A.6, the same calculation holds up to logarithmic factors in $K$ and $1/\delta$.

### A.2.4. OPTIMAL NON-UNIFORM ALLOCATION

Uniform allocation is suboptimal when anchors have different noise levels or task difficulties. The following result characterizes the optimal allocation for minimizing the worst anchor-estimation term.

**Proposition A.7** (Optimal non-uniform allocation). *Fix anchor locations $\{p_i\}_{i=1}^K$. Suppose the estimation error at anchor $i$ is bounded by*

$$\frac{C\sigma_i}{\sqrt{n_i}}, \tag{40}$$

*where $\sigma_i \geq 0$ and $\sum_{i=1}^K n_i = N$. If sample counts are allowed to be real-valued and at least one $\sigma_i$ is nonzero, the allocation minimizing*

$$\max_{1 \leq i \leq K} \frac{C\sigma_i}{\sqrt{n_i}} \tag{41}$$

*is*

$$n_i^\star = N \frac{\sigma_i^2}{\sum_{j=1}^K \sigma_j^2}. \tag{42}$$

*The achieved worst-anchor estimation error is*

$$C\sqrt{\frac{\sum_{j=1}^K \sigma_j^2}{N}}. \tag{43}$$

*Integer-valued allocations achieve the same rate up to rounding effects.*

*Proof.* Let $\lambda$ denote the value of the maximum. If

$$\max_i \frac{\sigma_i}{\sqrt{n_i}} \leq \lambda, \tag{44}$$

then necessarily $n_i \geq \sigma_i^2/\lambda^2$ for every $i$. Summing over $i$ gives

$$N = \sum_i n_i \geq \frac{1}{\lambda^2} \sum_i \sigma_i^2, \tag{45}$$

so every feasible allocation satisfies

$$\lambda \geq \sqrt{\frac{\sum_i \sigma_i^2}{N}}. \tag{46}$$

The allocation (42) attains equality for every $i$ with $\sigma_i > 0$, since

$$\frac{\sigma_i}{\sqrt{n_i^\star}} = \sqrt{\frac{\sum_j \sigma_j^2}{N}}. \tag{47}$$

Multiplying by $C$ gives (43). $\qquad\square$

**Implication.** The optimal allocation is proportional to variance proxy $\sigma_i^2$, not to standard deviation $\sigma_i$. Anchors with higher stochasticity require super-linear replication because the objective controls the worst anchor error.

A.2.5. BEYOND NEAREST-ANCHOR SURROGATES: STABLE INTERPOLATORS

The nearest-anchor analysis is conservative but does not automatically upper-bound an arbitrary neural network. A neural interpolator can be better or worse depending on optimization, regularization, representation, and extrapolation behavior. The following proposition states a sufficient condition under which the same type of decomposition applies.

**Assumption A.8** (Predictor stability in task parameter). For a realized predictor $\hat{f}$, there exists $\widehat{L} < \infty$ such that for all $z, t$ and all $p, p' \in \mathcal{P}$,

$$\|\hat{f}(z, p, t) - \hat{f}(z, p', t)\| \le \widehat{L}\|p - p'\|. \tag{48}$$

**Proposition A.9** (Coverage bound for stable interpolators). *Suppose Assumptions A.1 and A.8 hold. Then for any evaluation distribution $\nu$ over $(z, t)$,*

$$\sup_{p \in \mathcal{P}} \mathbb{E}_{\nu,\xi}\big[\|\hat{f}(z, p, t) - f^\star(z, p, t)\|\big] \le \max_i \mathbb{E}_{\nu,\xi}\big[\|\hat{f}(z, p_i, t) - f^\star(z, p_i, t)\|\big] + (L + \widehat{L})h_K. \tag{49}$$

*If $\widehat{L}$ is random, the same bound holds with $\mathbb{E}[\widehat{L}]h_K$ after taking expectation, provided the stability condition holds almost surely.*

*Proof.* For any $p$, let $i(p)$ be a nearest anchor. By the triangle inequality,

$$\|\hat{f}(z, p, t) - f^\star(z, p, t)\| \le \|\hat{f}(z, p_{i(p)}, t) - f^\star(z, p_{i(p)}, t)\| \tag{50}$$
$$+ \|\hat{f}(z, p, t) - \hat{f}(z, p_{i(p)}, t)\| \tag{51}$$
$$+ \|f^\star(z, p, t) - f^\star(z, p_{i(p)}, t)\|. \tag{52}$$

The last two terms are bounded by $\widehat{L}h_K$ and $Lh_K$, respectively. Taking expectations and the supremum over $p$ proves the result. $\square$

**Consequence for neural policies.** Equation (49) is the appropriate formal way to relate neural interpolation to the coverage–density argument. Neural networks may have favorable empirical interpolation behavior, but an improved coverage term requires explicit smoothness or stability control. In particular, Lipschitz regularity of $f^\star$ alone supports an $O(h_K)$ worst-case coverage term; an $O(h_K^2)$ term requires stronger smoothness and an interpolation procedure with second-order approximation accuracy.

More generally, if an interpolator satisfies a deterministic approximation bound of order $\beta > 0$,

$$\sup_{p \in \mathcal{P}} \|\hat{f}(z, p, t) - f^\star(z, p, t)\| \lesssim \max_i \|\hat{f}(z, p_i, t) - f^\star(z, p_i, t)\| + Bh_K^\beta, \tag{53}$$

then the uniform-allocation trade-off becomes

$$\mathcal{E}_\beta(K) \lesssim C\sigma\sqrt{\frac{K}{N}} + BK^{-\beta/d}, \tag{54}$$

which is optimized at

$$K_\beta^\star \asymp \left(\frac{B^2 N}{\sigma^2}\right)^{\frac{d}{d+2\beta}}, \qquad \mathcal{E}_\beta(K_\beta^\star) \asymp \sigma^{\frac{2\beta}{d+2\beta}} B^{\frac{d}{d+2\beta}} N^{-\frac{\beta}{d+2\beta}}. \tag{55}$$

For the nearest-anchor/Lipschitz case, $\beta = 1$. A second-order rate $\beta = 2$ is possible only under stronger smoothness assumptions, such as bounded second derivatives in $p$, together with an interpolator that actually realizes second-order approximation.

A.2.6. ROBUSTNESS TO DISCONTINUITIES

Worst-case and average-case guarantees behave differently when the task map has discontinuities. A discontinuous region of small measure cannot improve a supremum bound: a worst-case query may still fall exactly in that region. Therefore measure-dependent robustness statements must be formulated for distributional risk.

Let $\mu$ be a test distribution over $p \in \mathcal{P}$. Suppose $\mathcal{P}$ decomposes into a smooth region $\mathcal{P}_{\mathrm{sm}}$ and an exceptional region $\mathcal{P}_{\mathrm{ex}}$ such that

$$\mu(\mathcal{P}_{\mathrm{ex}}) \leq \epsilon. \tag{56}$$

Assume $f^{\star}$ is $L$-Lipschitz on $\mathcal{P}_{\mathrm{sm}}$, the anchors cover $\mathcal{P}_{\mathrm{sm}}$ with fill distance $h_{\mathrm{sm}}$, and the loss is bounded on the exceptional region by $M$:

$$\mathbb{E}_{\nu,\xi}\left[\|\hat{f}(z,p,t) - f^{\star}(z,p,t)\|\right] \leq M, \qquad p \in \mathcal{P}_{\mathrm{ex}}. \tag{57}$$

Then

$$\mathbb{E}_{p\sim\mu,(z,t)\sim\nu,\xi}\left[\|\hat{f}(z,p,t) - f^{\star}(z,p,t)\|\right] \leq \max_i \mathbb{E}_{\nu,\xi}\left[\|\hat{f}(z,p_i,t) - f^{\star}(z,p_i,t)\|\right] + Lh_{\mathrm{sm}} + \epsilon M. \tag{58}$$

This bound is distributional. Without the expectation over $p \sim \mu$, the term $\epsilon M$ cannot be used to control the supremum over all task parameters.

**Practical implication.** For contact-rich manipulation, discontinuities are best handled by phase segmentation, explicit mode-conditioned models, or additional anchors near suspected mode boundaries. The theory supports these interventions because they either reduce the smooth-region fill distance or reduce the probability mass and loss magnitude associated with exceptional regions.

### A.2.7. HEAVY-TAILED NOISE

The sub-Gaussian assumption is convenient but may fail in robotics because rare catastrophic executions can produce heavy-tailed errors. The following statement gives a conservative expectation bound for the ordinary sample mean under finite-moment assumptions.

**Proposition A.10** (Sample mean under finite moments). *Let $\varepsilon_1, \ldots, \varepsilon_n$ be independent, mean-zero scalar random variables satisfying*

$$\mathbb{E}|\varepsilon_j|^q \leq \sigma^q \tag{59}$$

*for some $q \in (1, 2]$. Then*

$$\mathbb{E}\left|\frac{1}{n}\sum_{j=1}^n \varepsilon_j\right| \leq 2^{1/q}\sigma n^{-(1-1/q)}. \tag{60}$$

*In particular, $q = 2$ recovers the usual $n^{-1/2}$ expectation rate, while the rate degrades as $q \downarrow 1$.*

*Proof.* By Jensen's inequality,

$$\mathbb{E}\left|\frac{1}{n}\sum_{j=1}^n \varepsilon_j\right| \leq \left(\mathbb{E}\left|\frac{1}{n}\sum_{j=1}^n \varepsilon_j\right|^q\right)^{1/q}. \tag{61}$$

For $q \in (1, 2]$, the von Bahr–Esseen inequality gives

$$\mathbb{E}\left|\sum_{j=1}^n \varepsilon_j\right|^q \leq 2\sum_{j=1}^n \mathbb{E}|\varepsilon_j|^q \leq 2n\sigma^q. \tag{62}$$

Therefore

$$\left(\mathbb{E}\left|\frac{1}{n}\sum_{j=1}^n \varepsilon_j\right|^q\right)^{1/q} \leq 2^{1/q}\sigma n^{1/q-1}, \tag{63}$$

which proves (60). $\qquad\square$

**High-probability robustness.** Under heavy-tailed noise, the ordinary sample mean generally does not satisfy the sub-Gaussian high-probability bound in Lemma A.5. If high-probability guarantees are required under only finite variance, the anchor estimator should be replaced by a robust mean estimator such as median-of-means or Catoni's estimator. Such estimators can recover $O(\sigma\sqrt{\log(1/\delta)/n})$ deviations under finite-variance assumptions, up to universal constants, but this requires changing the estimator.

A.2.8. TWO-STAGE CORE–BOUNDARY ALLOCATION

We next formalize the intuition behind a two-stage protocol. Unlike the main decomposition, this result depends on a stylized regional model. It should be interpreted as guidance for budget design, not as a claim that the empirical training pipeline exactly solves the corresponding optimization problem.

Suppose $\mathcal{P}$ is partitioned into two regions,

$$\mathcal{P} = \mathcal{P}_{\mathrm{core}} \cup \mathcal{P}_{\mathrm{bd}}, \qquad \mathcal{P}_{\mathrm{core}} \cap \mathcal{P}_{\mathrm{bd}} = \emptyset. \tag{64}$$

Let $v_r$ denote the $d$-dimensional volume of region $r \in \{\mathrm{core}, \mathrm{bd}\}$, and let $\sigma_r$ denote its noise scale. Assume region-specific anchor sets satisfy

$$h_r(K_r) \leq c_r \left(\frac{v_r}{K_r}\right)^{1/d}, \tag{65}$$

and that $N_r$ samples are allocated to region $r$, uniformly over $K_r$ anchors. The regional error model is

$$\mathcal{E}_r(K_r, N_r) \leq C\sigma_r \sqrt{\frac{K_r}{N_r}} + Lc_r \left(\frac{v_r}{K_r}\right)^{1/d}. \tag{66}$$

**Proposition A.11** (Regional budget split under worst-case balancing)**.** *Under the regional model* (66)*, optimizing $K_r$ within each region gives*

$$\mathcal{E}_r^\star(N_r) \asymp (C\sigma_r)^{\frac{2}{d+2}} (Lc_r v_r^{1/d})^{\frac{d}{d+2}} N_r^{-\frac{1}{d+2}}. \tag{67}$$

*If the objective is to minimize the maximum of the two regional errors, the balanced split satisfies*

$$\frac{N_{\mathrm{core}}}{N_{\mathrm{bd}}} \asymp \frac{\sigma_{\mathrm{core}}^2 c_{\mathrm{core}}^d v_{\mathrm{core}}}{\sigma_{\mathrm{bd}}^2 c_{\mathrm{bd}}^d v_{\mathrm{bd}}}. \tag{68}$$

*Proof.* For fixed $N_r$, minimizing (66) over $K_r$ is the same calculation as in (37), with $Lc_{\mathcal{P}}$ replaced by $Lc_r v_r^{1/d}$ and $\sigma$ replaced by $\sigma_r$. This yields (67). To minimize the maximum regional error, the two optimized errors should be balanced unless one region receives a boundary allocation. Setting $\mathcal{E}_{\mathrm{core}}^\star(N_{\mathrm{core}}) \asymp \mathcal{E}_{\mathrm{bd}}^\star(N_{\mathrm{bd}})$ and raising both sides to the power $d + 2$ gives

$$\frac{(C\sigma_{\mathrm{core}})^2 (Lc_{\mathrm{core}} v_{\mathrm{core}}^{1/d})^d}{N_{\mathrm{core}}} \asymp \frac{(C\sigma_{\mathrm{bd}})^2 (Lc_{\mathrm{bd}} v_{\mathrm{bd}}^{1/d})^d}{N_{\mathrm{bd}}}. \tag{69}$$

Canceling common constants gives (68). □

**Interpretation.** For a worst-case objective, the split depends on region volume, geometry, and noise scale. Larger or geometrically harder regions require more coverage; noisier regions require more replication. If $c_{\mathrm{core}} \approx c_{\mathrm{bd}}$, the split simplifies to

$$\frac{N_{\mathrm{core}}}{N_{\mathrm{bd}}} \asymp \frac{v_{\mathrm{core}}}{v_{\mathrm{bd}}} \left(\frac{\sigma_{\mathrm{core}}}{\sigma_{\mathrm{bd}}}\right)^2. \tag{70}$$

Thus, a boundary region with larger noise can require a non-negligible fraction of the budget even if its volume is smaller. If the objective is average-case performance under a task distribution, the probability masses of the two regions also enter the allocation; the worst-case split above should then be modified accordingly.

A.2.9. RELATION TO STANDARD NONPARAMETRIC RATES

The rate in (39) is consistent with standard nonparametric regression over Lipschitz/Hölder classes. For a $d$-dimensional Lipschitz function class, the minimax integrated squared-error rate is of order

$$N^{-\frac{2}{d+2}}, \tag{71}$$

which corresponds to root mean-squared error of order

$$N^{-\frac{1}{d+2}}. \tag{72}$$

Our bound has the same exponent for the nearest-anchor Lipschitz case. The analysis here differs from classical non-parametric regression in that the design is summarized by anchor coverage and per-anchor replication, which makes the coverage–density trade-off explicit and directly interpretable for robotic adaptation.

A.2.10. CONNECTION TO NEURAL AND TRANSFORMER POLICIES

The formal results above do not require the deployed policy to be a nearest-neighbor rule. They instead provide a baseline decomposition that exposes what any successful interpolating policy must manage: anchor accuracy and task-space coverage.

For neural networks, the relevant question is whether training induces a stable interpolator in the task parameter. If the learned predictor has controlled Lipschitz constant in $p$, Proposition A.9 applies. If the task map is smoother than Lipschitz and the model realizes a higher-order approximation property, the generic $\beta$-order calculation predicts that fewer anchors may be required. However, such improvements are conditional on smoothness, optimization, and regularization; they are not implied by neural-network parametrization alone.

For transformer-based policies, attention can sometimes be interpreted as a learned kernel smoother over context examples or demonstrations. This interpretation is useful as intuition, but it does not by itself imply a coverage guarantee. A transformer may reduce interpolation error when its learned similarity metric aligns with the task geometry, but it may also extrapolate poorly outside the support of the adaptation data. Therefore, the formal theory should be read as an anchor-based statistical baseline, while the empirical results test whether the learned policy inherits the predicted qualitative behavior.

A.2.11. LIMITATIONS OF THE ANALYSIS

The analysis is intentionally conservative. Its guarantees are most informative when the following conditions approximately hold:

1. The task parameter $p$ captures the dominant source of variation during adaptation.

2. The target map varies smoothly in $p$ over the region being analyzed.

3. Anchor-level estimation error can be controlled through repeated or sufficiently similar samples.

4. The evaluation distribution over $(z, t)$ is covered by the data used to estimate anchor predictors, or a separate common-support assumption is justified.

The guarantees may be loose or inapplicable when contact-mode discontinuities dominate, when state distributions shift substantially across task parameters, when failures induce heavy-tailed noise and the estimator is not robust, or when the neural policy has uncontrolled extrapolation behavior. These limitations motivate the practical design choices used in the method: phase-aware data collection, denser sampling near difficult regions, robust aggregation when failures are present, and a staged expansion from high-density core tasks to boundary tasks.

A.2.12. SUMMARY

The theory yields four main takeaways.

1. **Coverage–density decomposition.** Under Lipschitz regularity and a nearest-anchor surrogate, the error separates into an anchor-estimation term and a coverage term $Lh_K$.

2. **Interior optimum.** With a fixed budget $N$, increasing the number of anchors improves coverage but worsens per-anchor estimation. This yields an optimal scaling $K^\star \asymp N^{d/(d+2)}$ in the Lipschitz case.

3. **Noise-aware allocation.** When anchor noise levels differ, worst-case estimation error is minimized by allocating samples proportional to $\sigma_i^2$, not $\sigma_i$.

4. **Conditional extensions.** Stable neural interpolators satisfy analogous bounds with an explicit predictor-stability term. Higher-order coverage rates require stronger smoothness and approximation assumptions; they should not be claimed solely from using neural networks or transformers.

