# OpenReview forum: "Escaping the Diversity Trap in Robotic Manipulation via Anchor-Centric Adaptation"
_ICML.cc/2026/Conference — ICML 2026 regular_

### Official Review · Reviewer_kzk2 · 2026-02-19

**Soundness:** 3
**Presentation:** 3
**Significance:** 3
**Originality:** 4
**Overall Recommendation:** 4
**Confidence:** 4

**Summary:**

This paper first observed and formulated the tradeoff between coverage (extrapolation) and density (estimation). Building on the theoretical analysis, it proposed ACA, a budget-allocation framework for fine-tuning VLAs.

**Compliance With Llm Reviewing Policy:**

Affirmed.

**Final Justification:**

My concerns have been adequately addressed, and I tend not to change my score.

**Key Questions For Authors:**

Is there any link between Assumption 2 to the learning capability of models? (Since the error bound decreases inversely to the number of samples)

**Limitations:**

See weaknesses 2-3.

**Strengths And Weaknesses:**

### Strengths:
1. The paper proposed a theoretical analysis of the trade-off and leveraged its result to propose ACA.
2. The paper clearly stated the assumptions made and why they are necessary.
3. The proposed spatial metric, S@1~S@3, provides a deeper understanding of model performance.



### Weaknesses:
1. The assumptions might seem necessary in the theoretical analysis, but whether they hold in the real world is not verifiable, especially A3.
2. Only one base VLA family model (pi 0.5 and 0) is tested in this work.
3. The authors should also test some (naive) fine-tuning-based baselines as a fair comparison to ACA. If not possible, the paper should state why such a comparison is unfair.

---

> ### Author Rebuttal · Authors · 2026-03-31
>
> ## Response to R3 kzk2
> **We thank the reviewer for recognizing the paper's theoretical clarity, explicit assumption design, and informative spatial evaluation.** We welcome the opportunity to address your concerns and improve the quality of our work.
>
> Notation: **W** = Weakness; **Q** = Question
>
> >### W1: Assumption Validity
>
> 1. **A3 is a conservative upper bound, not an empirical claim.** Its role is to make the coverage (extrapolation) term explicit in the error bound. Real neural policies interpolate more smoothly than the nearest-neighbor surrogate — Appendix A.1.9 analyzes this formally, showing improved $O(h^2)$ bias scaling for neural interpolators (Thm. A.10) and robustness under heavy-tailed noise (Prop. A.6). The bound is therefore conservative by designl; the qualitative predictions (interior optimum, two‑stage benefit) transfer as lower bounds on achievable performance.
>
> 2. **Empirical validation of the proxy.** We further evaluate the practical proxy used by ACA. We measured the relation between the teacher-forced deviation score $e(p)$ (Eq. 16) and actual difficulty across all 4 tasks (40 probe conditions total, 10 per task, N=100 setting):
>    |Correlation|Value|
>    |-|-:|
>    |Spearman($e(p)$, failure rate)|0.72|
>    |Pearson($e(p)$, open-loop error)|0.65|
>
>    These correlations show that $e(p)$ reliably identifies high-risk boundary regimes in practice, which is exactly the mechanism that A3 is designed to capture.
>
> >### W2: Model Generalizability
>
> 1. **Additional architectures.** Under $N=100$, ACA yields **consistent gains across three architectures**:
>
>    |Setting|$\pi_{0.5}$|OpenVLA-OFT|FLOWER|
>    |-|:-:|:-:|:-:|
>    |Baseline|31.7%|24.6%|17.1%|
>    |**+ ACA (ours)**|**72.5%**|**59.6%**|**45.0%**|
>    |$\Delta$|+40.8%|+35.0%|+27.9%|
>
> 2. **Analysis.** ACA's anchor‑centric principle operates at the data allocation level, independent of model architecture, and delivers substantial improvements across all three.
>
> >### W3: Fine‑Tuning Baselines
>
> 1. **Main Baseline.** The $\pi_{0.5}$ baseline in Table 1 is already **standard full-parameter VLA fine-tuning**; the difference is its **data collection strategy** (maximal-diversity sampling, with each demonstration collected at a distinct condition). Thus, ACA is already compared against the standard fine-tuning protocol under the same trajectory budget.
>
> 2. **Ablation on update design.** We further isolate the roles of **error-driven mining** and the **residual update** in Table 2. This includes a non-residual variant corresponding to standard boundary expansion without the constrained update:
>
>    |Mining|Residual|Success|
>    |:-:|:-:|:-:|
>    |✗|✗|57.1%|
>    |✗|✓|58.8%|
>    |✓|✗|64.2%|
>    |✓|✓|**72.5**%|
>
>    The gain comes from combining targeted boundary data with a stable update mechanism.
>
> 3. **Additional data‑collection baselines.** To further address the concern, we added three stronger baselines under identical conditions ($\pi_{0.5}$, $N\in\{50,100\}$):
>    - **Uncertainty-driven**: max-coverage Stage 1, then variance-based Stage 2 collection;
>    - **DAgger-style**: max-coverage Stage 1, then human-takeover correction with replay buffer mixing;
>    - **Curriculum**: core-region (S@1 & S@2) Stage 1, then random boundary (S@3) expansion with full-parameter fine-tuning.
>
>    |Method|$N=50$|$N=100$|
>    |-|:-:|:-:|
>    |Baseline (diverse)|13.8%|31.7%|
>    |Uncertainty‑driven|15.0%|35.4%|
>    |DAgger‑style|18.3%|39.2%|
>    |Curriculum|23.3%|42.5%|
>    |**ACA (ours)**|**46.3%**|**72.5%**|
>
>    ACA substantially outperforms the alternatives, demonstrating that the gain stems from structured data allocation—suppressing estimation variance at the core before expanding coverage—not from a particular fine‑tuning mechanism.
>
> >### Q1 — Relation Between A2 and Model Learning Capability
>
> 1. **A2 captures the effect of data density, rather than model capacity.** Assumption A2,
>     $$
>     \mathbb{E}\|\hat{f}(z,p_i,t) - f^\star(z,p_i,t)\| \leq C\sigma/\sqrt{n_i},
>     $$
>
>     describes how repeated supervision at the same condition reduces local estimation variance. The $1/\sqrt{n_i}$ dependence reflects a standard variance-shrinking effect of repeated noisy supervision, and is therefore primarily a property of the **data regime**.
>
> 2. **Model capability enters through the approximation term.** In our neural-network extension (Appendix A.1.9), the total error is separated into an **estimation** term (governed by $N$, $K$, $\sigma$) and an **approximation** term (governed by model capacity and fill distance $h$). Stronger models can lower the approximation floor and improve architecture-dependent constants, but they do **not remove the need for sufficient per-condition density**.
>
> 3. **Connection to the empirical results**. This also explains Fig. 7. When the base model is stronger (e.g., $\pi_{0.5}$ vs. $\pi_0$), improved data allocation can be converted more effectively into performance, leading to larger absolute gains under ACA.

---

> > ### Author Rebuttal · Reviewer_kzk2 · 2026-04-02
> >
> > Thank you for the response. Can you also report the Spearman and Pearson p-values, or any other statistics besides the correlation coefficients?

---

> > > ### Author Response · Authors · 2026-04-03
> > >
> > > **We thank the reviewer for the helpful follow-up and continued engagement.** We provide the additional statistics below.
> > >
> > > ---
> > >
> > > **Results.** The correlations are computed over **40 probe conditions** across all 4 tasks (**10 probe conditions per task**, $N=100$ setting).
> > >
> > > | Metric          | Correlation Target         | Value |  $p$-value |       95% CI |
> > > | --------------- | -------------------------- | :----: | :---------: | :-----------: |
> > > | Spearman $\rho$ | $e(p)$ vs. failure rate    |  0.72 | < 0.0001 | [0.53, 0.84] |
> > > | Pearson $r$     | $e(p)$ vs. open-loop error |  0.65 | < 0.0001 | [0.42, 0.80] |
> > >
> > > **Analysis.** Both correlations are statistically significant ($p$ < 0.001) with strictly positive 95% confidence intervals, indicating that $e(p)$ is meaningfully aligned with actual rollout difficulty. Since ACA uses $e(p)$ as a **selection proxy** to rank and identify high-risk boundary regimes, rather than as a precise predictor, this level of correlation is sufficient to support its practical use.
> > >
> > > ---
> > >
> > > Please let us know if any further clarification would be helpful. We sincerely appreciate your constructive feedback!

---

### Official Review · Reviewer_bnFb · 2026-02-21

**Soundness:** 2
**Presentation:** 3
**Significance:** 2
**Originality:** 3
**Overall Recommendation:** 5
**Confidence:** 4

**Summary:**

This work proposes Anchor-Centric Adaptation (ACA) for efficiently adapting robot policies by reducing the number of required demonstrations. Instead of sparsely collecting single demonstrations for a vide state space, ACA collects repeated demonstrations for a small number of "anchor" states, which then get expanded at key boundary states to increase coverage.

This approach is theoretically motivated and evaluated via four real-robot experiments and several ablation studies.

**Compliance With Llm Reviewing Policy:**

Affirmed.

**Final Justification:**

The author's rebuttals to all the reviewers addressed my concerns. With the added experiments and analysis I feel confident that this work makes a valuable contribution to reducing demonstration collection efforts for robot learning. I therefore recommend to accept this work.

**Key Questions For Authors:**

1. How does ACA perform with a different, non $\pi0$-based VLA architecture? See Weaknesses for examples of possible candidate models. Further experiments would demonstrate the generalizability of the proposed method and I would raise my score accordingly.
2. How does this work relate to Hg-Dagger?
3. Is the proposed method limited to flow matching policies in cartesian action spaces. How would the authors expect this approach to perform on alternative models such as ACT, and on other action spaces such as Joint Control?
2. It is unclear to me how the data budget is used. Adding this information would help. Is my understanding correct: For example, with N=100, 70 demonstrations would be used on 7 anchors with 10 demonstrations each. Then 10 "mining demonstrations" would be collected to identify the policy boundaries. The remaining 20 demonstrations in the budget would be used to collect 5 demonstrations for each of the 4 worst performing boundary states. Please clarify.

**Limitations:**

I encourage the authors to discuss, whether the focus of ACA on flow-matching based policies is an inherent limitation. Similarly, this approach seems currently limited to applications with clearly defined "Degrees of Freedom", where anchor states can be easily defined, such as object placement.

**Strengths And Weaknesses:**

## Strengths
- Well presented and easy to understand.
- Well motivated and addresses the important topic of data efficient robot learning.
- Strong theoretical grounding in the formalization of Coverage-Density Tradeoff
- Introduction of a relevant evaluation metric Success@X, measuring success rates at different work space sizes.
- Experiments show a clear improvement over the baseline experiments with the various design choices evaluated via ablation experiments.

## Weaknesses
- Limited Evaluation regarding Models and Tasks. While four real-robot experiments are presented, three of them are similar pick-and-place tasks, which might explain the highly similar success rates across these three tasks as presented in Tab 1.
    - **Models**. The current evaluation focuses only on one architecture ($\pi0$).
Testing should extend to other VLA models, e.g. OpenVLA (Kim 2024), FLOWER (Reuss 2025), SPEAR-1 (Nikolov 2025) or GR00T (Bjorck 2025) to show generalizability.
Especially FLOWER and SPEAR-1 are already designed for low-data regimes and might further benefit from ACA.
    - **Simulation Tasks**: While real-robot experiments are of critical importance in robot learning, this work would further be strengthened by leveraging the advantages of simulated tasks. In simulated environments such as LIBERO or Robocasa, the task diversity can be easily increased to encompass different manipulation objects, background scenes and more complex tasks. In this settings, it might not always be feasible to define good anchor states without overfitting to a small number of manipulation objects. Simulation experiments could show that ACA is not limited to highly controlled laboratory settings.
    - **Real World Tasks**: A task like cloth folding with not as easily discretized states such as cube position might provide additional insights.
    - **Non Cartesian Experiments**: The method seems constrained to work with cartesian object poses and cartesian action outputs by the VLA model. Experimenting with Joint Based control would demonstrate wider applicability.
- **Alternate approaches**: This work seems related to approaches like DAgger (Ross 2011), Hg-Dagger (Kelly 2019) and its successors, or curriculum learning approaches. Discussions and even experiments would further strengthen this work.




### Minor Comments:
- Tab 1. is difficult to parse due to missing separation between the grouped columns. The rest of the manuscript, especially the various Figures, are *exceptionally* well executed.
- I believe Stage 1 / Stage 2 refers to the training phases, where the policy is further refined. I encourage the authors to consider describing their framework as a 3-Step approach, as depicted in Figure 2: Anchoring Step. Mining Step. Coverage Step. I personally believe this might further improve clarity, and fit to the stages of Data Collection.

---

> ### Author Rebuttal · Authors · 2026-03-31
>
> ## Response to R2 bnFb
> **We thank the reviewer for recognizing our paper's clarity, strong theoretical grounding, and experimental design.** We appreciate the opportunity to address the concerns below.
>
> Notation: **W** = Weakness; **Q** = Question
>
> >### W1 & Q1: Model Generalizability
>
> 1. **Results**. We evaluated across three architectures under the same budget $N=100$. ACA yields **consistent gains** over the max-coverage baseline:
>    |Setting|$\pi_{0.5}$|OpenVLA-OFT|FLOWER|
>    |-|:-:|:-:|:-:|
>    |Baseline|31.7%|24.6%|17.1%|
>    |**+ACA**|**72.5**%|**59.6**%|**45.0**%|
>    |$\Delta$|+ 40.8%|+ 35.0%|+ 27.9%|
>
> 2. **Gains scale with pretraining quality**. Stronger pretrained representations provide a more stable vector field prior, **amplifying the variance-reduction effect of anchor densification**. FLOWER shows smaller gains, which is plausibly related to its lighter pretraining setup, as it was designed for efficient VLA training; nonetheless, ACA still yields a +27.9% gain. Our Coverage–Density bound (§3) is generic; the interior optimum holds across architectures under non‑vanishing noise.
>
> >### W2: Simulation Experiments
>
> 1. **Results**. We applied ACA on LIBERO (first 3 tasks from the Spatial suite, $N=100$ per task, joint training) while varying the anchor ratio:
>    |Anchor Ratio|Success|
>    |-|:-:|
>    |0% (Baseline)|30.0%|
>    |30%|51.7%|
>    |50%|**73.3**%|
>    |70%|65.0%|
>
> 2. **Transferability.** ACA remains effective in simulation, demonstrating that its benefits are **not limited to our physical lab setup** and generalize across scenes and object variations.
>
> 3. **Cross‑domain insight.** The optimal anchor ratio is lower in simulation (50%) than on real robots (~75%). This aligns with our theory: simulation has **lower effective noise** $\sigma$ due to near‑deterministic dynamics, and the optimal allocation shifts toward more coverage. This difference thus serves as an **additional empirical validation** of the Coverage–-Density framework.
>
> >### W3: Additional Real-World Task (Cloth Folding)
>
> 1. **Results.** We added a cloth-folding task, where states are less discretizable than rigid object placement:
>    |Setting (20 rollouts)|$N=50$|$N=100$|
>    |-|:-:|:-:|
>    |Baseline|15%|25%|
>    |ACA|**40**%|**60**%|
>
> 2. **Analysis.** ACA does **not** require rigid objects or Cartesian grids. It needs a **task-relevant condition space** over which anchors can be defined. For cloth folding, anchors are defined over **canonical configurations**, parameterized by the cloth's **initial center**, **orientation**, and **grasp-start region**. This provides dense support on representative folding conditions, after which ACA expands to harder boundary cases. Crucially, these need not be precisely specified: our random-placement experiments (please refer to **R1 (MSVZ)-W3**) show that even random workspace-spanning positions suffice.
>
> >### W4 & Q3: Non-Cartesian Action Spaces
>
> 1. **Joint-control results.** We replaced Cartesian delta actions with **joint-velocity control** on $\pi_{0.5}$. ACA remains effective, demonstrating that its benefits are **not tied to Cartesian action spaces**:
>    |Setting|$N=50$|$N=100$|
>    |-|:-:|:-:|
>    |Baseline|10.8%|23.8%|
>    |ACA|**42.1**%|**62.1**%|
>
> 2. **Theoretical generality.** Our Coverage-Density analysis (§3) characterizes error in terms of **per-condition sampling density**, independent of the specific action parameterization. The interior optimum therefore holds for joint‑control spaces under non‑negligible noise. More broadly, the core insight — insufficient density per condition increases estimation variance — extends beyond flow matching to other policy classes trained under data scarcity.
>
> >### W5 & Q2: Alternate Data Approaches
>
> 1. **Results.** Please refer to our response to **R1 (MSVZ)-W1**, ACA outperforms the baselines under identical budgets.
>
>    |Method|$N=50$|$N=100$|S@3 ($N=100$)
>    |-|:-:|:-:|:-:|
>    |Baseline|13.8%|31.7%|12.5%|
>    |Uncertainty‑driven|15.0%|35.4%|18.8%|
>    |DAgger‑style|18.3%|39.2%|21.3%|
>    |Curriculum|23.3%|42.5%|27.5%|
>    |**ACA**|**46.3**%|**72.5**%|**65.0**%|
>
> 2. **Relation to Hg-Dagger.** HG-DAgger operates in an **online interaction loop** where the expert intervenes during policy rollouts to provide corrections. ACA addresses a structurally different setting: **batch collection under a fixed small budget**, where all demonstrations are pre-collected. Under extreme budget constraints, reactive corrections remain local and cannot compensate for an unstable core policy. ACA first stabilizes the core via repeated anchors, then targets high-risk boundaries — directly addressing the budget allocation problem.
>
> >### W6 & W7 & Q4: Presentation and Budget
> 1. **We sincerely thank the reviewer for the presentation suggestions.** We will revise Table 1 formatting and adopt the three-step description (Anchoring → Mining → Coverage).
> 2. For the **exact budget breakdown** in the main table, please refer to **R1 (MSVZ)-Q2**.

---

> > ### Author Rebuttal · Reviewer_bnFb · 2026-04-01
> >
> > I thank the authors for their detailed response to all the reviews.
> > With the newly added experiments my concerns are addressed, and I will raise my score accordingly.
> >
> > Regarding the Demonstration Budgets:
> > I believe there might be an interesting analysis in how much the Stage 2 "refinement" improves over the Stage 1 policy, i.e.  $N_A = 70$ vs $N_A + N_{probe} + N_{bd} = 100$
> > I encourage the authors to consider adding such an analysis as to the appendix.

---

> > > ### Author Response · Authors · 2026-04-01
> > >
> > > **We sincerely thank you for your continued engagement and for maintaining a positive assessment of our work. We are glad to hear that our responses have effectively addressed your concerns.**
> > >
> > > We also appreciate the suggestion regarding the demonstration budget breakdown. We agree that analyzing the incremental contribution of **Stage 2 refinement** over the **Stage 1 anchor policy** is informative. In particular, comparing the performance of the policy trained with only $N_A=70$ anchor demonstrations against the full ACA pipeline with $N_A + N_{\text{probe}} + N_{bd}=100$ can help clarify how much additional gain comes from the boundary-mining and residual-adaptation stage.
> > >
> > > We will include this analysis in the appendix. Concretely, we will report the Stage-1-only policy and its improvement after Stage 2 (and Stage 3) under the same total budget, so that the contribution of the refinement stage is made explicit.

---

### Official Review · Reviewer_MSVZ · 2026-03-12

**Soundness:** 3
**Presentation:** 2
**Significance:** 2
**Originality:** 3
**Overall Recommendation:** 5
**Confidence:** 4

**Summary:**

This paper addresses the problem of data-efficient real-robot adaptation of Vision-Language-Action (VLA) models. The central claim is that the standard heuristic of "maximizing coverage" by collecting diverse, single-shot demonstrations is counterproductive under tight budgets — a phenomenon the authors term the diversity trap. The work makes three contributions: (1) an empirical demonstration of an inverted-U performance trend with respect to the number of unique conditions; (2) a theoretical Coverage–Density Trade-off framework that decomposes worst-case policy error into estimation (density) and extrapolation (coverage) terms, yielding an interior optimal anchor count $K^*$; and (3) Anchor-Centric Adaptation (ACA), a two-stage practical pipeline that first consolidates a stable policy skeleton via repeated anchor demonstrations, then expands to high-risk boundary conditions through teacher-forced error mining and LoRA-based residual adaptation. Experiments on a Franka Panda robot across four tabletop tasks demonstrate consistent gains over a diversity-first baseline under budgets of 50–150 trajectories.

**Compliance With Llm Reviewing Policy:**

Affirmed.

**Final Justification:**

My questions are totally solved.

**Key Questions For Authors:**

1. Anchor initialization: How are the initial anchors chosen in practice? Can the anchors be chosen automatically and how sensitive is performance to suboptimal initial anchor placement beyond the four spatial layouts in Fig. 4?
2. Budget: In the main results (Table 1), what is the budget breakdown, i.e. What are $N_A$, $N_{probe}$, and $N_{bd}$?

**Limitations:**

Lack of comparisons with strong baselines with additional data collection designs. Pipeline complexity is high.

**Strengths And Weaknesses:**

## Strengths
1. Well-motivated research. The question of how to allocate a scarce demonstration budget is directly important for real-world VLA deployment, and the "maximize diversity" heuristic is genuinely prevalent in the community. The inverted-U empirical trend is a crisp, reproducible finding that challenges common intuition.
2. Real-robot validation. Conducting all experiments on physical hardware rather than simulation is commendable. The non-vanishing noise assumption ($\sigma > 0$), which is the crux of the theoretical argument, is only meaningful in real hardware settings.

---
## Weaknesses
1. Lack of Comparisons. The empirical validation only proves that ACA is better than a "naive" baseline $\pi_{0.5}$ where no additional data collection strategies are applied. By failing to compare ACA against the active learning, continual learning, self-corrected, uncertainty-driven and robust augmentation data collection methods, the authors have not demonstrated that ACA is the optimal solution to the diversity trap among existing state-of-the-art techniques.
2. Pipeline Complexity: ACA's data collection procedure is fundamentally a human-in-the-loop, stop-and-go pipeline that requires the operator to repeatedly pause and wait between stages. It would always be hard to deploy into broad use with such human-in-the-loop complexity. Yet the paper uses total trajectory count $N$ as the sole cost metric, ignoring the non-trivial wall-clock time and human effort introduced by these interruptions. Since the baseline collects $N$ trajectories continuously with no algorithmic intervention, the comparison is conducted under asymmetric true costs, potentially overstating ACA's practical advantage.
3. Anchor placement relies on human domain knowledge. The anchors are described as forming "a coarse, quasi-uniform cover of the reachable workspace." In practice, this requires the operator to know the task-relevant workspace geometry beforehand and to place anchors manually. The paper does not describe an automated or principled procedure for anchor selection. This limits the method's applicability to settings where workspace structure is known a priori, which is often not the case in novel deployments. The claimed practical contribution is thereby narrower than presented.

---

> ### Author Rebuttal · Authors · 2026-03-31
>
> ## Response to R1 MSVZ
>
> **We thank the reviewer for the constructive feedback and recognition of our problem formulation and the real-robot validation.** We address the concerns below.
>
> Notation: **W** = Weakness; **Q** = Question
>
> >### W1: Comparisons with Data-collection Baselines
>
> 1. **Setup.** We add three baselines under identical conditions ($\pi_{0.5}$, $N\in\{50,100\}$, same two-stage budget split).  Following each method's standard protocol, all three use diverse single-shot sampling in Stage 1.
>    - **Uncertainty-driven**: Diverse Stage 1 seed policy; Stage 2 collects at conditions with highest flow-matching variance.
>    - **DAgger-style**: Diverse Stage 1 seed policy; Stage 2 aggregates human corrections at rollout failure states via replay mixing.
>    - **Curriculum**: Collects from core regions first (S@1–S@2); expands to boundary (S@3).
>
> 2. **Results**. All three improve over the naïve baseline, but **ACA remains substantially better**.
>     |Method|$N=50$|$N=100$|S@3 ($N=100$)|
>     |-|:-:|:-:|:-:|
>     |Baseline (Max-Coverage)|13.8%|31.7%|12.5%|
>     |**Uncertainty‑driven**|15.0%|35.4%|18.8%|
>     |**DAgger‑style**|18.3%|39.2%|21.3%|
>     |**Curriculum**|23.3%|42.5%|27.5%|
>     |**ACA (ours)**|**46.3**%|**72.5**%|**65.0**%|
>
> 3. **Analysis**. The S@3 gap reveals the core issue: all three baselines inherit the diversity trap from their single-shot Stage 1, producing an unstable policy skeleton that subsequent expansion cannot rescue. Curriculum shares ACA's core→boundary structure but lacks (1) **anchor repetition** to suppress estimation noise and (2) **deviation-driven boundary selection** to target where extrapolation error dominates. ACA is the only method that explicitly addresses both the density and coverage terms in our decomposition (Prop. 3.1).
>
> >### W2: Pipeline Complexity and Practical Cost
>
> 1. **Dominant cost.** In real-robot adaptation, the primary bottleneck is **human teleoperation data**, whose collection time scales directly with the number of demonstrations $N$. GPU training and deviation screening between stages are fully automated and require no operator presence. The practical cost of a method is therefore best evaluated by the total operator effort required to reach a deployable success rate.
>
> 2. **Cost to reach a usable policy.** Using the 70% success threshold as a reference, ACA achieves this at $N=100$, while the max‑coverage baseline requires $N\approx 550$:
>
>    |Setting|$N$|Success|Human Time|Total Time|
>    |-|:-:|:-:|:-:|:-:|
>    |ACA|100|72.5%|~4 h|~15 h|
>    |Baseline (Max-Coverage)|550|70.8%|~20 h|~38 h|
>
>    ACA's human effort is **5× lower** (4 h vs. 20 h), and total time (including training and screening) is also substantially lower. Thus, despite a more structured pipeline, ACA **reduces the overall deployment burden**.
>
> 3. **Operational efficiency.** ACA is **stage‑wise** rather than per‑trajectory interactive—the operator does not pause after every rollout. Probe trajectories are **reused** as supervision, so no collected data is wasted. The additional overhead is limited to offline training / screening between stages, which is automated and small relative to teleoperation time.
>
> >### W3 & Q1: Anchor Initialization and Sensitivity
>
> 1. **Density over absolute position**. Our theoretical framework (Prop. 3.1) dictates that the fundamental driver of ACA's stability is **local density** (repeated sampling to suppress estimation noise), **rather than absolute spatial precision**. The structured layouts in Fig. 4 were chosen for visual clarity and controlled analysis, not as a prerequisite.
>
> 2. **Robustness to Zero-Prior (Random) Placement**. To quantify sensitivity beyond Fig. 4, we evaluate two additional configurations where 6 anchor positions are sampled uniformly at random within the workspace ($N=100$).
>    |Setting|Success|
>    |-|:-:|
>    |No Anchors (Baseline)|31.7%|
>    |Random Placement 1|70.4%|
>    |Random Placement 2|68.8%|
>    |Center Rect (Fig. 4a)|72.5%|
>
> 3. **Summary**. Both random configurations substantially outperform the baseline and closely match the best structured layout, confirming that **the key driver of ACA's gains is repeated sampling density, not precise anchor geometry**. Practically, this **eliminates the need for layout engineering or automated selectors**; operators can simply choose a few diverse, workspace-spanning positions, demonstrating ACA's strong robustness to heuristic anchor placement.
>
> >### Q2: Budget Breakdown
>
> 1. **Main-table Allocation**. The precise allocation for Table 1 (6 anchors) is:
>
>    |Setting|$N_A$|$N_{\text{probe}}$|$N_{bd}$|
>    |-|:-:|:-:|:-:|
>    |$N=50$|35|5|10|
>    |$N=100$|70|10|20|
>    |$N=150$|110|15|25|
>
> 2. **Example ($N=100$)**. 70 anchor trajectories across 6 anchors(10~12 per anchor); 10 probe trajectories for deviation scoring; the top-3 highest-deviation boundary conditions are selected, around which 20 additional trajectories are collected for Stage 2 residual adaptation.

---

> > ### Author Rebuttal · Reviewer_MSVZ · 2026-04-07
> >
> > I appreciate author providing those additional results that mostly addressed my concerns.
> >
> > However, I think the paper missed a critical ablation: Stage 1 (anchor) + Stage 2 with randomly selected locations for additional data collection (instead of selecting top-k) + residual adapter. Without this comparison, it remains unclear whether the performance gain in Stage 2 comes from the error-driven selection criterion, or merely from additional data coverage at the boundary regions.

---

> > > ### Author Response · Authors · 2026-04-07
> > >
> > > >### Response — Ablation of Error-Driven Selection
> > >
> > > We thank the reviewer for the positive acknowledgment and for this precise follow-up.
> > >
> > > We would like to draw attention to **Table 2 (§5.3)** in the main paper, which **already contains this exact comparison**:
> > >
> > > | Stage 1 (Anchor) | Boundary Selection | Residual | Mean Acc. (%) |
> > > |:-:|---|:-:|--:|
> > > | ✓ | Random | ✓ | 58.8 |
> > > | ✓ | **Error-driven (top-$k$)** | ✓ | **72.5** |
> > >
> > > The row **Mining ✗, Residual ✓** corresponds precisely to: Stage 1 anchor stabilization + **randomly selected** boundary locations + residual adapter. The **+13.7% gap** directly isolates the contribution of error-driven mining. This confirms that the Stage 2 gain does not arise merely from additional boundary coverage, but from **targeting the highest‑risk regions** identified by teacher‑forced deviation. With the same budget and residual architecture, random expansion recovers less than half of ACA’s Stage 2 improvement.
> > >
> > > We hope this clarifies that the contribution of boundary mining is already isolated in our existing ablation.

---

### Decision · Program_Chairs · 2026-04-30

**Decision:**

Accept (regular)

**Comment:**

This paper is a solid contribution to ICML 2026 based on the AC's evaluation. The reviewers evals (5,4,5) after rebuttal.

It proposes Anchor-Centric Adaptation, an explore-exploit algorithm with theoretical analysis to tackle the diversity of robot learning tasks.
The authors showed good experimental results with reasonable analysis.
The figure illustrations and content presentation is clearly above the average level of ICML papers in previous years.

There are still weaknesses over model usage, simulation and real world tasks. But in general it is a good paper from the AC's perspective.